# Impacts and uncertainties of climate-induced changes in watershed inputs on estuarine hypoxia

**Authors**: Kyle E. Hinson[1], Marjorie A.M. Friedrichs[1], Raymond G. Najjar[2]; Maria Herrmann[2], Zihao Bian[3], Gopal Bhatt[4,5], Pierre St-Laurent [1], Hanqin Tian[6], Gary Shenk[7,5]

[1]Virginia Institute of Marine Science, William & Mary, Gloucester Point, VA 23062, USA
[2]Department of Meteorology and Atmospheric Science, The Pennsylvania State University, University Park, PA 16802, USA
[3]International Center for Climate and Global Change, Auburn University, Auburn, AL 36849, USA
[4]Department of Civil & Environmental Engineering, The Pennsylvania State University, State College, 16801, USA
[5]United States Environmental Protection Agency Chesapeake Bay Program Office, Annapolis, 21401, USA
[6]Schiller Institute for Integrated Science and Society, Department of Earth and Environmental Sciences, Boston College, Chestnut Hill, MA 02467, USA
[7]U.S. Geological Survey, Virginia/West Virginia Water Science Center, Richmond, VA 23228, USA

*Correspondence to*: Kyle E. Hinson (kehinson@vims.edu; kyle.e.hinson@gmail.com)

## Abstract

Multiple climate-driven stressors, including warming and increased nutrient delivery, are exacerbating hypoxia in coastal marine environments. Within coastal watersheds, environmental managers are particularly interested in climate impacts on terrestrial processes, which may undermine the efficacy of management actions designed to reduce eutrophication and consequent low-oxygen conditions in receiving coastal waters. However, substantial uncertainty accompanies the application of Earth System Model (ESM) projections to a regional modeling framework when quantifying future changes to estuarine hypoxia due to climate change. In this study, two downscaling methods are applied to multiple ESMs and used to force two independent watershed models for Chesapeake Bay, a large coastal-plain estuary of the eastern United States. The projected watershed changes are then used to force a coupled 3-D hydrodynamic-biogeochemical estuarine model to project climate impacts on hypoxia, with particular emphasis on projection uncertainties. Results indicate that all three factors (ESM, downscaling method, and watershed model) are found to contribute substantially to the uncertainty associated with future hypoxia, with the choice of ESM being the largest contributor. Overall, in the absence of management actions, there is a high likelihood that climate change impacts on the watershed will expand low-oxygen conditions by 2050, relative to a 1990s baseline period; however, the projected increase in hypoxia is quite small (4%) because only climate-induced changes in watershed inputs are considered and not those on the estuary itself. Results also demonstrate that the attainment of established nutrient reduction targets will reduce annual hypoxia by about 50% compared to the 1990s. Given these estimates, it is virtually

certain that fully implemented management actions reducing excess nutrient loadings will
outweigh hypoxia increases driven by climate-induced changes in terrestrial runoff.
**Short Summary**
Climate impacts are essential for environmental managers to consider when implementing
nutrient reduction plans designed to reduce hypoxia. This work highlights relative sources of
uncertainty in modeling regional climate impacts on the Chesapeake Bay watershed and
consequent declines in Bay oxygen levels. The results demonstrate that planned water quality
improvement goals are capable of reducing hypoxia levels by half, offsetting climate-driven
impacts to terrestrial runoff.

## 1 Introduction

Over the past several decades, estuarine and coastal ecosystems have been subject to elevated levels of hypoxia relative to the open ocean (Gilbert et al., 2010), and are anticipated to be affected by multiple climate change impacts including terrestrial runoff changes (Breitburg et al., 2018) and rising temperatures (Whitney, 2022). Increases in precipitation volume and intensity are likely to increase streamflow and associated nutrient and sediment export to coastal systems (Howarth et al., 2006; Lee et al., 2016; Sinha et al., 2017). Rising atmospheric temperatures will increase soil temperatures and alter evapotranspiration, soil biogeochemical cycling and plant responses (Schaefer and Alber, 2007; Wolkovich et al., 2012; Ator et al., 2022), also affecting riverine nutrient export to marine habitats. Further changes to agricultural practices driven by these same climate impacts are also likely to contribute to altered nutrient applications and subsequent soil cycling (Wagena et al., 2018). Altogether, climate impacts in the terrestrial environment may further eutrophy coastal ecosystems (Najjar et al., 2010), altering the phenology and biogeochemical rates of nutrient consumption and exacerbating hypoxia (Testa et al., 2018).

Future estimates of coastal hypoxia have increased substantially over the past decade, likely influenced by increased access to biogeochemical modeling tools and regional climate projections needed for finer scale modeling and analyses (Fennel et al., 2019). The majority of coastal hypoxia climate impact studies have focused on a select few coastal locations including the Baltic Sea (Meier et al., 2011a,b; Meier et al., 2012; Neumann et al., 2012; Ryabchenko et al., 2016; Saraiva et al., 2019a,b; Wåhlström et al., 2020; Meier et al., 2021; Meier et al., 2022), Chesapeake Bay (Wang et al., 2017; Irby et al., 2018; Ni et al., 2019; Testa et al., 2021; Tian et al., 2021; Cai et al., 2021), and the Gulf of Mexico (Justić et al., 1996; Justić et al., 2007; Lehrter et al., 2017; Laurent et al., 2018). Other projected changes to dissolved oxygen ($O_2$) levels have been documented in nearshore environments including the North Sea (Meire et al., 2013; Wakelin et al., 2020), Arabian Sea (Lachkar et al., 2019), California Current System (Dussin et al., 2019; Siedlecki et al., 2021; Pozo Buil et al., 2021), and coastal waters surrounding China (Hong et al., 2020; Yau et al., 2020; Zhang et al., 2021; Zhang et al., 2022). Hypoxia projections in relatively smaller estuaries have also been documented in the Elbe (Hein et al., 2018), Garonne (Lajaunie-Salla et al., 2018), and Long Island Sound (Whitney and Vlahos, 2021).

Broadly speaking, these climate impact studies apply either a range of idealized changes to conduct a sensitivity study or utilize long-term projections derived from Earth System Models (ESMs) (IPCC, 2013). When directly applying such projections to study regional coastal oxygen responses, dynamically or statistically downscaled estimates of atmospheric and marine variables are typically used to continuously simulate climate impacts or to calculate and apply a change factor (Carter et al., 1994; Anandhi et al., 2011) to a shorter historical time period. Quantifying the relative uncertainties from various sources including ESM, downscaling methodology, internal variability, and hydrological model is not new to the field of climate research (Hawkins and Sutton, 2009; Yip et al., 2011; Northrop and Chandler, 2014) or watershed applications (Bosshard et al., 2013; Vetter et al., 2017; Wang et al., 2020; Ohn et al., 2021). Questions of uncertainty due to climate effects in past marine ecosystem impact studies have often been addressed by selecting some combination of ESMs and/or emissions scenarios (Meier et al., 2011a; Ni et al., 2019; Saraiva et al., 2019b; Meier et al., 2019; Meier et al., 2021; Pozo Buil et al., 2021). Additionally, some studies have also sought to account for the importance of managed nutrient runoff from terrestrial (Irby et al., 2018; Saraiva et al., 2019a; Bartosova et al., 2019;

Pihlainen et al., 2020) and atmospheric (Yau et al., 2020; Meier et al., 2021) sources and their
impacts on oxygen levels. Despite some comprehensive efforts to identify sources of uncertainty
in coastal oxygen projections (Meier et al., 2019; 2021), few studies have evaluated uncertainties
introduced by the choice of specific downscaling method and/or terrestrial model. These factors
represent additional sources of variability when estimating future hypoxia and are inherent in
regional simulations of coastal dynamics.
The Chesapeake Bay, which is the largest estuary in the continental United States (Kemp et
al., 2005), has undergone intensive management efforts to improve water quality and oxygen
levels over the past three decades. These management efforts have focused on the reduction of
excess nitrogen, phosphorus, and sediment loadings to the Bay (USEPA, 2010) and continuous
adaptive monitoring efforts to evaluate progress in restoring water quality (Tango and Batiuk,
2016). Recent analyses of monitoring data have demonstrated improvements in water quality
throughout the Bay despite the trajectory of recovery being slowed by extreme weather events
(Zhang et al., 2018). Observed lags in water quality responses to nutrient reductions (Murphy et
al., 2022) are also evident in recent years (Zahran et al. 2022). Despite the difficulties in
assessing long-term improvements in water quality due to strong interannual variability, new
research has demonstrated that the Chesapeake Bay is more resilient to recent and ongoing
climate change impacts that have decreased oxygen levels as a result of decades of nutrient load
reductions (Frankel et al., 2022).
In recent years managers have recognized the importance of investigating whether the
originally established Total Maximum Daily Loads (USEPA, 2010) will need to be adjusted to
ensure the attainment of water quality standards for the Chesapeake Bay as the climate changes
(Chesapeake Bay Program, 2020; Hood et al., 2021). Increasing temperatures and precipitation
are anticipated to affect watershed snowpack, soil moisture levels, terrestrial nutrient cycling,
and associated streamflow, streamflow generation, and flooding (Shenk et al., 2021b), potentially
altering the efficacy of nutrient reduction strategies. Increases in nutrient and carbon inputs to the
Bay resulting from climate change and anthropogenic stressors have already been documented
over the course of the past century (Pan et al., 2021; Yao et al., 2021), and are anticipated to
increase in the 21$^{st}$ century as well (Wang et al., 2017; Irby et al., 2018; Ni et al., 2019). For
example, Irby et al. (2018) directly tested the role of future nutrient reductions via a sensitivity
analysis of mid-century climate effects and found substantial alleviation of hypoxic conditions
when management targets were met, despite significantly increasing water temperatures.
However, that study applied spatially constant changes in watershed inputs derived from a
specific watershed model, one downscaling technique and a median estimate of ESM
projections. A more robust effort to produce a range of scenarios incorporating multiple
watershed models, downscaling techniques and ESMs is needed to assess uncertainty estimates
of projected hypoxia, which can be used to guide decision making that explicitly considers what
levels of environmental risk are acceptable for Chesapeake Bay stakeholders.
The present study applies multiple downscaled ESMs to two independently developed
watershed models with significantly different representation of watershed processes and spatial
scale; both are used to force a coupled hydrodynamic-biogeochemical estuarine model in order
to better constrain the relative uncertainties of future terrestrial runoff estimates on estuarine
hypoxia (Shenk et al., 2021a). The resulting ensemble of numerical experiments includes
realistic climate forcings and an extensive set of regional linked watershed-estuarine
deterministic model simulations. The framework established in this research assesses the relative
uncertainties introduced by choice of ESM, downscaling methodology, and regionally focused
watershed model in quantifying changes to $O_2$ levels in the estuary. Additionally, this
investigation constrains the bounds of changes to Chesapeake Bay hypoxia (defined herein as $O_2$
$< 2$ mg $L^{-1}$) with and without the effects of management actions, using an ensemble of realistic
watershed forcings. The study provides a roadmap for environmental managers to design climate
impact assessments that are better able to quantify the range of possible future levels of hypoxia,
which can be influenced by nutrient management actions.
**2 Methods**
**2.1 Monitoring data**
Monthly estimates of freshwater streamflow, inorganic nitrogen, and organic nitrogen at the
non-tidal monitoring stations nearest the head of tide of the three largest tributaries to the
Chesapeake Bay (Susquehanna, Potomac, and James; Fig. 1a; Table S1) were used to evaluate
the performance of watershed models. Streamflow and nitrogen load estimates are derived from
observations that are collected at U.S. Geological Survey (USGS) stream gages (U.S. Geological
Survey, 2022) and comprise part of the USGS River Input Monitoring program in the
Chesapeake Bay watershed (Mason and Soroka, 2022). Estimates for the nitrogen species were
calculated using a weighted statistical regression process that accounts for the variability
introduced by time, discharge, and season (Hirsch et al., 2010).
Main stem bay observations collected over the period 1991-2000, accessible via a data
repository maintained by the Chesapeake Bay Program (CBP; Olson 2012; CBP DataHub 2022),
were used to assess estuarine model skill (see Sect. 2.2). Since 1984, numerous water quality
data have been collected along the Bay's main stem and throughout its tributaries at semi-
monthly to monthly intervals as part of the Water Quality Monitoring Program. These data were
collected at the surface, above and below the pycnocline, and at the bottom for chemical
variables including nitrate and organic nitrogen, and throughout the entire water column at 1-2 m
intervals for $O_2$. Twenty CBP stations were selected for model comparison at the surface and
bottom (Fig. 1b, Table S2), including those most frequently sampled and those located along the
entirety of the Bay's main channel where hypoxia commonly occurs (Officer et al., 1984; Hagy
et al., 2004). Estimates of annual hypoxic volume (AHV), defined as the volume of hypoxic
water integrated over the year (with units of volume*time), were taken from the Bever et al.
(2013; 2018; 2021) interpolation of $O_2$ measurements at 56 CBP stations.
**2.2 Estuarine and watershed modeling tools and evaluation**
Model simulations are conducted with ChesROMS-ECB, a fully coupled, three-dimensional,
hydrodynamic and estuarine carbon biogeochemistry (ECB) implementation of the Regional
Ocean Modeling System (ROMS; Shchepetkin and McWilliams 2005) developed for the
Chesapeake Bay (Xu et al., 2011) with 20 terrain-following vertical levels and an average
horizontal resolution of approximately 1.8 kilometers in the estuary's main stem (Feng et al.,
2015; St-Laurent et al., 2020; Frankel et al., 2022). Two parameter changes were recently made
to improve the representation of modeled oxygen: (1) a decrease of the maximum growth rate of
phytoplankton, which, following Irby et al. (2018), preserves the temperature-dependent linear
$Q_{10}$ described in Lomas et al. (2002), and (2) a decrease in the critical bottom shear stress from
0.010 Pa to 0.007 Pa, which increases the resuspension of organic matter and is well within the
range of observed shear stresses evaluated by Peterson (1999).
Estimates of watershed streamflow, nitrogen loading, and sediment loading to drive the
estuarine model were obtained via two independently developed models of the Chesapeake Bay
watershed: the Dynamic Land Ecosystem Model (DLEM; Yang et al., 2015; Yao et al., 2021)
and the USEPA Chesapeake Bay Program's regulatory Phase 6 Watershed Model (Phase 6;
Chesapeake Bay Program, 2020). Both models were applied to generate comparable reference
runs over the average hydrology period of 1991-2000, chosen because it reflects the decade used
by the Chesapeake Bay Program to calculate Total Maximum Daily Loads (USEPA, 2010) and
assess water quality improvements. Outputs from both watershed models were aggregated into
10 major river input locations (RIM in Fig. 1). Watershed outputs were mapped to estuarine
variables as in Frankel et al. (2022), except that a more realistic partitioning of terrestrial organic
nitrogen loading into labile and refractory pools was implemented such that the percent
refractory organic nitrogen loading increases with streamflow at high flow volumes (Appendix
A).
Atmospheric conditions, including temperature and winds, were obtained from the ERA5
reanalysis dataset (C3S, 2017) as in Hinson et al. (2021). Coastal boundary conditions were
interpolated to match the nearest physical and nutrient observations, as in previous work (Da et
al., 2021). In order to isolate the impacts of climate-driven changes in watershed inputs,
atmospheric and coastal boundary conditions were kept the same in all model simulations under
realistic 1991-2000 conditions, for both reference runs (1991-2000) and all future scenarios
211  (2046-2055).
Watershed and estuarine model skill was evaluated by comparing results from the two
reference scenarios to available data (see Sect. 2.1). Nash–Sutcliffe efficiencies (Nash and
Sutcliffe, 1970) were used to evaluate watershed model performance of freshwater streamflow
and nutrient loadings. Estuarine model skill was evaluated by comparing model outputs
matching the spatio–temporal variability of observations at 20 main stem stations over the 10-
year reference period. Average bias (model output minus observed value) and root-mean squared
difference (RMSD) of annual $O_2$, nitrate ($NO_3$), and dissolved organic nitrogen (DON)
concentrations were calculated at the surface and bottom of the water column. AHV estimates
were calculated by summing the daily volume of model cells containing low-oxygen waters ($O_2$
$< 2$ mg $L^{-1}$) and are expressed in units of $km^3$ d following Bever et al. (2013; 2018; 2021). Daily
net primary production estimates were integrated over the entire water column and averaged
across the Bay and month before being compared to average Bay-wide estimates from Harding et
al. (2002).
**2.3 Projected changes in atmospheric temperature and precipitation**
Mid-21st century projected changes in atmospheric temperature and precipitation under a
high emissions scenario (RCP 8.5; Cubasch et al., 2013) were obtained for multiple ESMs from
the 5th Coupled Model Intercomparison Project (CMIP5) that were regionally downscaled via
two statistical methodologies: Multivariate Adapted Constructed Analogs (MACA; Abatzoglou
and Brown, 2012; downloaded from MACAv2-METDATA) and Bias-Correction and Spatial
Disaggregation (BCSD; Wood et al., 2004; downloaded from Reclamation, 2013). (Note that
downscaled CMIP5 ESM output was used because downscaled CMIP6 ESM output was not yet
available when the research began.) Downscaled MACA and BCSD projections have an average
spatial resolution of approximately 0.042° and 0.125°, respectively. A delta approach
(Prudhomme et al., 2002; Anandhi et al., 2011) was used to estimate the absolute change in
atmospheric temperature and fractional change in precipitation over the Chesapeake Bay
watershed. In this delta approach (also commonly referred to as a perturbation method or
change-factor method), the difference in a given climate variable (i.e., air temperature or
precipitation) is calculated by first subtracting monthly downscaled ESM estimates averaged
over a hindcast period (in this case 1981-2010) from average monthly future projections (in this
case 2036-2065). The resulting mean annual cycle (with monthly resolution) in the delta (i.e., the
absolute change in temperature or fractional change in precipitation) is then applied to reference
atmospheric forcing inputs (in this case for 1991-2000) to generate future watershed scenarios
(in this case for 2046-2055, hereafter referred to as mid-century) and limit uncertainty introduced
by interannual variability. An additional step to modify precipitation intensity is also included in
all climate scenarios, following the methodology outlined in Shenk et al. (2021b). Thirty-year
averaging periods were used to limit potential biases introduced by multidecadal oscillations.
To reduce the computational load of applying the dozens of available ESMs to our combined
watershed-estuarine modeling framework for a full factorial experiment, the Katsavounidis-Kuo-
Zhang (KKZ; Katsavounidis et al., 1994) algorithm was applied to select a subset of five ESMs
from both downscaled datasets. KKZ is an objective procedure for selecting a subset of members
that best span the spread of the full ensemble in a multivariate space. Because changes to
hypoxia must be computed after a subset of ESMs are selected, the downscaled results were
classified in terms of changes to the two variables most likely to influence hypoxia: air
temperature from May–October (i.e., the historic hypoxic season in Chesapeake Bay) and
precipitation from November–June (corresponding to the highest set of correlation coefficients
when regressed against historical AHV estimates; Supplementary Material, Fig. S1). The KKZ
algorithm first selected an ESM nearest to the center of the cluster of models in the two-
parameter space, which is referred to hereafter as the Center ESM, before iteratively selecting
additional ESMs that were furthest from the center of the distribution and other previously
selected ESMs (Fig. 2, Table S3). The next four selected ESMs are referred to as Hot/Wet,
Cool/Wet, Hot/Dry, and Cool/Dry ESMs to denote whether they are cooler, hotter, wetter, or
drier, relative to the Center ESM. The specific ESMs selected based on MACA and BCSD differ
slightly; however, three of the five models are the same (Cool/Dry, Hot/Dry, and Cool/Wet). The
selection process incrementally adds members to those previously selected, so that the entire
ensemble is ordered and a subset of any size can be selected. This method has proven effective at
covering the largest range of outcomes using the fewest ESMs in watersheds across the United
States in previous research (Ross and Najjar, 2019). This ESM selection process allows for a
more robust comparison of the distribution of ESMs from multiple downscaled datasets as
opposed to individual ESM comparisons that may privilege one downscaling method over
others. However, because inexact matches among ESMs can impact the quantification of relative
uncertainty (Sect. 2.5), additional scenarios were simulated as needed for the Center and
Hot/Wet ESMs, which were different for the two downscaling techniques (Fig. 2, Table S3).
Future change in temperature and precipitation between the two downscaling methods are shown
for the Center ESM (Fig. 3); changes for the other four ESMs are found in the Supplementary
Material (Fig. S2).
**2.4 Experiments**
Three numerical experiments (sets of simulations) were conducted to evaluate the impacts of
climate scenario factors, management conditions, and the use of a subset of ESMs on future
AHV projections and uncertainty (Table 1). To isolate climate impacts on AHV from the
watershed alone, direct atmospheric and oceanic forcings to the Bay were held the same as in the
reference simulations (see Sect. 2.3) for all experiments. The first experiment (Multi-Factor)
evaluates the relative change in AHV (hereafter defined as ΔAHV) between the 1991-2000 and
2046-2055 time periods due to the following factors: ESM, downscaling method, and watershed
model (Table 1, Fig. 4). Atmospheric deltas from ten downscaled ESMs (five from MACA and
five from BCSD) were applied directly to the two watershed models for a total of 20 simulations.
A separate Phase 6 climate-reference run is used to evaluate the impacts of climate alone by
holding land use and nutrient applications constant. This differs slightly from the Phase 6
reference run that simulates realistic and interannually varying nutrient inputs and terrestrial
conditions and is compared against observations (Sect. 2.2). Two additional simulations were
conducted with Phase 6 to account for the fact that the ESMs selected by the KKZ method were
not identical for MACA and BCSD (Table 1, Fig. 2).
The second experiment (Management) applied the same deltas used for Phase 6 MACA
scenarios in the Multi-Factor experiment (thereby varying runoff and nutrient loading), but also
included the effect of changing environmental management conditions (affecting nutrient inputs
to and export from the terrestrial environment), for a total of five additional simulations. These
Management simulations assume that reduction targets for nutrient and sediment runoff are met
in accordance with established management goals (USEPA, 2010). One additional scenario was
conducted in which management goals were imposed and climate change was not.
The third experiment (All ESMs) applied all 20 MACA downscaled ESM deltas to the
DLEM scenarios without any changes to management conditions, thereby only modifying
changes in runoff and nutrient export without intentional nutrient reductions, for a total of 20
additional simulations. Comparing the results of the first (Multi-Factor) and third (All ESMs)
experiments highlights the strengths and limitations of using a subset of ESMs.
**2.5 Climate scenario analyses**
To analyze climate impacts on Chesapeake Bay hypoxia, changes in $O_2$ and AHV were
compared between the reference runs and the future simulations. Relative $O_2$ impacts introduced
by the three climate scenario factors (ESM, downscaling method, and watershed model) were
determined by applying an analysis of variance (ANOVA) approach to average ΔAHV estimates
for each climate scenario. This method has been previously applied to the quantification of
uncertainty sources in climate and hydrological applications (Hawkins and Sutton, 2009; Yip et
al., 2011; Bosshard et al., 2013; Ohn et al., 2021). To use this method in this study, an average
annual metric is first calculated for an outcome of interest (i.e., change in streamflow, nitrogen
loading, or hypoxic volume) within the Multi-Factor experiment. Then, the relative uncertainty is
determined by calculating the sum of squares due to individual effects for each experimental
factor (ESM, downscaling method, or watershed model). Following Ohn et al. (2021), the
cumulative uncertainty is quantified for successive uncertainties introduced by each factor as
well as their interactions, removing the unexplained interaction term described in Bosshard et al.
(2013). The two additional ESM scenarios described previously (Table 1, Table S3) were used
due to the inexact matches between MACA and BCSD ESMs selected by KKZ. Despite five
ESMs being used in combination with only two downscaling methods and two watershed models
in this analysis, the approach outlined (Bosshard et al., 2013; Ohn et al., 2021) accounts for this
factor imbalance (five vs. two) by repeatedly subsampling combinations of two ESM scenarios
from the five available. An example of this methodological approach is described in Appendix B.
Relative frequency histograms and cumulative distributions were used to quantify the overall
likelihoods of increasing/decreasing ΔAHV across the entire range of future scenarios. Average

changes in the spatial distribution of $O_2$ over the typical hypoxia season (May–September) were compared among all climate scenarios with no changes to management conditions. Results were considered significant if at least 80% of model scenarios tested agree on the direction of $O_2$ change in the estuary, as in Tebaldi et al. (2011).

## 3 Results

### 3.1 Model Skill

#### 3.1.1 Watershed Models

Modeled streamflow, nitrate loading, and organic nitrogen loading from the three largest Bay tributaries are comparable to observed monthly estimates derived from weighted statistical regressions (see Sect. 2.1). At the most downstream USGS streamgages on the Susquehanna, Potomac, and James Rivers, both Phase 6 and DLEM streamflow estimates have higher skill (Nash–Sutcliffe efficiencies closer to 1.0) relative to nitrate and organic nitrogen loading estimates (Table 2, Fig. S3). Although the overall skill of Phase 6 and DLEM is similar, Phase 6 generally exhibits higher model skill than DLEM in estimating monthly nitrate loading, while DLEM demonstrates greater skill in simulating organic nitrogen loading.

#### 3.1.2 Estuarine Model

The two reference simulations, forced with loadings from DLEM and Phase 6, demonstrate substantial skill in representing key main stem estuarine biogeochemical variables, including $O_2$, $NO_3$, DON, primary production, and AHV (Table 3) throughout the Bay's main stem. Overall, all modeled variables at the surface and bottom of the water column forced by both DLEM and Phase 6 lie within one standard deviation of observations. Modeled $O_2$ is slightly greater than spatio–temporally paired observations at the bottom, and slightly lower than observations at the surface throughout the entire year (Table 3) and in the summer period of hypoxia (Fig. 5a-b), leading to a bias that is relatively small compared to the standard deviations of observed $O_2$ concentrations across the entire year (Table 3). Additionally, modeled $O_2$ performs similarly to or better than the results included in the multi-model comparison presented in Irby et al. (2016). Modeled average annual $NO_3$ and DON are also within the range of observations at both the surface and bottom (Table 3). Whole Bay net primary production agrees well with observed estimates (Harding et al., 2002) reported over a similar time period (Table 3). Finally, modeled AHV compares favorably to data-derived interpolated estimates (Table 3; Fig. 5c), with increased hypoxia in wet years compared to dry years. Average AHV estimates using Phase 6 and DLEM inputs are, respectively, 16% and 26% greater than interpolated observations (Table 3; Fig. 5c) and approximately half the model estimates lie within the estimated uncertainties (RMS % error) of the interpolation methodology ($\pm$ 13%; Bever et al., 2018). Model estimates of AHV are generally slightly greater when ChesROMS-ECB is forced by DLEM watershed outputs as opposed to those from Phase 6 (Table 3; Fig. 5c).

### 3.2 Future (mid-21$^{st}$ century) projections of watershed streamflow and nutrient loading

Increasing temperatures and changing precipitation throughout the Bay watershed produce
different streamflow responses for the two watershed models. On average, Phase 6 climate
scenarios increase watershed runoff relative to the reference run by 4-6% while DLEM climate
scenarios decrease average flow by 1-4% (Table 4). The annual flow changes range from -12 to
+15% among ESM scenarios, with wetter ESMs tending to increase annual watershed
streamflow while drier ESM scenarios generally decrease average watershed runoff, with a lesser
impact due to atmospheric warming (Table 4; Fig. 6a). For both watershed models and
downscaling methods, the Cool/Wet ESM produces the greatest increase in annual streamflow.
Overall, the greatest variability in changes to streamflow estimates is due to ESM, as MACA and
BCSD scenarios increase or decrease annual streamflow by comparable amounts (Table 4; Fig
6a).
Chesapeake Bay Phase 6 watershed model climate scenarios increase average annual total
nitrogen (TN) by +30% and +45% for MACA and BCSD respectively, but do not substantially
change DLEM TN loads (+1% and -2% for MACA and BCSD, respectively; Fig. 7). Greater
Phase 6 TN loadings are primarily due to extreme values in the Cool/Wet climate scenarios and
are driven by increases in refractory DON (Fig. 7a). While DLEM scenarios show increases in
the percentage of inorganic nitrogen and labile organic forms of total nitrogen loads, the
contribution of particulate organic nitrogen (PON) decreases, resulting in little to no increases in
overall TN loading (Fig. 7a). Phase 6 produces wetter climate scenarios increasing TN loading
more than drier scenarios (Table 4; Fig 6b), with this effect being most pronounced for the
Cool/Wet ESM. Phase 6 also produces the greatest percent changes in the southern rivers (James,
York, Rappahannock), while DLEM produces similar percent changes in all rivers (Fig. 7b).
Some Phase 6 climate scenarios substantially increase the average loading change in smaller
watersheds like the Rappahannock and York, which increase TN between 77-172% and 32-
430%, respectively, and are comparable to the absolute change in Susquehanna TN loading (Fig.
7b). In contrast with the Multi-Factor experiment results, climate scenarios that include
management actions substantially reduce TN loading (-28%; Fig. 7, Table 4). Like other Phase 6
climate scenarios that do not account for management actions, the proportion of refractory
organic nitrogen increases for the climate scenarios with management (+49%), but in these cases
the average labile inorganic and organic nitrogen loadings also substantially decrease (-40%).
**3.3 Effects of future watershed change on estuarine $O_2$**
Climate change impacts on watershed streamflow and nitrogen loading substantially affect
estuarine hypoxia, even when, as in this study, direct climate effects on the Bay are not
considered. On average, the Multi-Factor climate scenarios decrease average summer bottom $O_2$
in the Bay's main stem while also slightly increasing $O_2$ at the surface in some mid-Bay areas
(Fig. 8). In the northern part of the main stem near the Susquehanna River outfall, model results
show consistent decreases in both bottom and surface summer $O_2$ (Fig. 8e,f). Further down the
main stem in the mid-Bay, surface $O_2$ increases in wet years, and experiences almost no change
in dry years (Fig. 8b,c). In the same region, bottom $O_2$ declines lessen during wet years and
worsen during dry years (Fig. 8e,f). Increasing $O_2$ levels are found in the shallow portions of the
major tidal tributaries (i.e., Potomac and James), but are more pronounced in wet years than dry
years (Fig. 8b-c,e-f). Altogether, average summer surface $O_2$ increases by $0.02 \pm 0.03$ mg L$^{-1}$
(average change and standard deviation) while bottom $O_2$ decreases by $0.03 \pm 0.06$ mg L$^{-1}$.

There are some clear distinctions in the overall changes to future AHV when evaluating all Multi-Factor experiments. Climate effects on the watershed in DLEM increase AHV more so than in Phase 6 (5.6% vs 3.1%, respectively), but the overall standard deviation of DLEM ΔAHV results are greater than those for Phase 6 (Table 5). Similarly, using MACA vs. BCSD results in greater changes in ΔAHV (4.8% vs. 3.9%), albeit this difference due to the choice of downscaling method is less than that due to the choice of watershed model. Depending on the choice of ESM, ΔAHV ranges between +0.9% (for the Cool/Dry ESM) to +8.3 % (for the Cool/Wet ESM) with the Center ESM producing intermediate results (+4.4 %). When comparing the impact of a particular ESM, wetter models tend to produce greater ΔAHV than drier scenarios (Fig. 6c), although interannual variability is still large. When climate scenarios are downscaled using different methodologies (either MACA or BCSD), average ΔAHVs have some notable differences, e.g., applying the Cool/Dry model to Phase 6 produces opposite average changes to hypoxia depending on downscaling method (Fig. 6c). Considering all possible combinations of scenarios, ESM average annual projected AHV spans a range of 921-939 $km^3$ d for Phase 6 and 1019-1049 $km^3$ d for DLEM, and four out of five of the climate scenarios in the Multi-Factor experiment project increases in average AHV (Table 4).

When the full distribution of Multi-Factor scenarios is evaluated, the average and standard deviation of these annual ΔAHV results are estimated to be 37 ± 64 $km^3$ d (4.4 ± 7.4%; Fig 9). Wetter ESMs (blues in Fig. 9a) are more likely to increase hypoxia compared to drier ESMs, despite differences in downscaling method or watershed model. The likelihoods of the Cool/Dry or Hot/Dry ESM increasing hypoxia are only 58% or 50%, respectively, but these chances are greater than 80% for the Center, Hot/Wet, and Cool/Wet ESMs (Fig. 9a). Altogether, the Multi-Factor experiment results in 72% of the runs increasing AHV when considering climate change impacts on terrestrial runoff (Fig. 9b). (Note, however, that this cannot technically be considered to be a statistical probability as the KKZ selection process used to generate our sample of climate scenarios is neither random nor independent.)

The All-ESMs experiment produces similar results to those obtained when only a subset of five ESMs is used. Specifically, ΔAHV increases by 6.3 ± 3.5% using only five KKZ-selected ESMs and by 9.6 ± 1.7% when using all 20 ESMs (Fig. 10a,b; Model IDs further defined in Table S3). The use of five KKZ-selected ESMs covers approximately 69% of the total range of all 20 ESMs (Fig. 10c). Despite more than 15,000 options to choose from when selecting five out of 20 ESMs, the subset selected in this work demonstrates an improved ability to outperform a random selection of five ESMs (Fig. 10c) and generates a useful range of hypoxia projections.

The results of the Management experiment demonstrate the substantial impact of future nutrient reductions on hypoxia, decreasing average ΔAHV by 50 ± 7% relative to the 1990s (ΔAHV = -438 ± 47 $km^3$ d; Table 4; Fig. 11). Because there is a linear relationship between ΔAHV computed with Phase 6 MACA scenarios including management actions (ΔAHV$_{mgmt}$) and those without (ΔAHV = 0.56 * ΔAHV$_{mgmt}$ – 262; $R^2$=0.59, Fig. S4), ΔAHV$_{mgmt}$ can be estimated for any scenario by applying this linear model to the non-management scenario distribution. In effect, this linear relationship demonstrates a similar magnitude of relative nutrient export to and consequent hypoxia within the estuary. The result is a decrease of approximately 417 ± 67 $km^3$ d among all scenarios, within the range of the management scenario subset obtained here by applying only MACA downscaled ESMs to Phase 6. As expected, hypoxia increases in the Management experiment when climate impacts are also included relative to the reference management scenario, specifically by 17.1 ± 34.8 $km^3$ d or 3.8 ± 7.8% (Table 4; Fig 6c).

**3.4 Contributions to Climate Scenario Uncertainty**

Applying an ANOVA approach (Ohn et al., 2021) to watershed streamflow, nutrient loadings, and ΔAHV within the Multi-Factor experiment reveals that the relative uncertainties introduced by the choice of ESM, downscaling method, and watershed model vary substantially (Fig. 12). The choice of ESM is the dominant factor affecting changes to watershed streamflow and nutrient loadings (Fig. 12a-c) and comprises 59-74% of the total uncertainty. The choice of watershed model is the next largest source of uncertainty, making up 17-34% of the total variance in watershed changes, while the downscaling method only contributes 3-14%. Uncertainty in projected organic nitrogen loadings is particularly affected by the choice of watershed model, overwhelming the variability introduced by downscaling method, and strongly affecting estimates of total nitrogen change. Unlike changes to watershed flow and loadings, the uncertainty of projected changes to hypoxia is much more evenly distributed among the three scenario factors: 40%, 25%, and 35%, for ESM, downscaling method, and watershed model respectively (Fig. 12d).

**4 Discussion**

**4.1 Uncertainty in Climate Scenario Projections**

Projected changes in watershed streamflow and nutrient delivery to the Chesapeake Bay produce modest increases in estuarine hypoxia, with medium confidence (Mastrandrea et al., 2010). Hypoxic volume has a high degree of interannual variability, and future hypoxia estimates are highly sensitive to the choice of ESM, downscaling method, and watershed model (Fig. 6c). Although certain factors (particularly ESM and greenhouse gas emissions scenarios; Meier et al., 2021) have previously been extensively evaluated in coastal systems with regards to future hypoxia, the results presented here also demonstrate the importance of terrestrial forcings on estuarine oxygen levels.

In this study, future changes in watershed streamflow, nitrogen loadings, and estuarine hypoxia are found to be highly dependent on the selection of a specific ESM (Fig. 12), comprising a majority of the total uncertainty in watershed runoff and the greatest fraction of total uncertainty for $O_2$ levels. When only the effect of ESM choice is considered (and downscaling and hydrological model options are not; Fig. 10), the average projected change in AHV using only three ESMs (often chosen to represent cool, median, and hot scenarios) has a greater standard error than the selection of five ESMs using KKZ in this study. Directly comparing results from the experiment that compared five ESMs, two downscaling methods, and two watershed models (Multi-Factor) versus that which only considered the impact of multiple ESMs (All ESMs) shows a substantial overlap in the range of projected ΔAHV. In addition, multiple ESMs downscaled with a single methodology and applied to one hydrological model produced meaningfully different estimates of ΔAHV than a more balanced approach (Fig. 11).

Inter-model variability among ESMs appears to contribute most substantially to differences in Bay watershed inputs, but the choice of downscaling methodology can also affect these projections. The BCSD (Wood et al., 2004) and MACA (Abatzoglou and Brown, 2012) downscaling methodologies used here employ different approaches to reduce historical ESM biases, impacting the variability of spatio–temporal watershed hydrologic and water quality responses. The ability to statistically downscale ESMs accurately depends on the spatially

coarser ESM's ability to simulate synoptic-scale (~1000 km) patterns and may still
underestimate the distributional tails of changes to temperature and precipitation. This increases
the importance of properly selecting a subset of ESMs (Abatzoglou and Brown, 2012).
Watershed model variability is caused by differences in the representation of processes that
affect streamflow, those controlling the fate and transport of nutrients from land and in rivers,
and lag times of groundwater transport. The two watershed models used here project
substantially different results in watershed streamflow and nitrogen delivery, even when the
same changes to meteorological forcings are applied (Fig. 6). DLEM projects no change or
decreases in streamflow for nearly all scenarios, as opposed to greater average increases in
streamflow for Phase 6 scenarios (Fig. 6a), likely driven by differences in the representation of
evapotranspiration. Explicit soil biogeochemical processes within DLEM increase nitrification
rates in warmer climate scenarios, producing higher nitrate loadings than Phase 6 despite
comparable streamflow changes (Fig. 6b). The greater total nitrogen loadings produced by Phase
6 are largely a consequence of its parameterizations for erosion and refractory nitrogen bound to
sediment. Increases in bioavailable nitrate loadings, unlike refractory organic nitrogen that
comprises the majority of DON loadings, produce greater levels of primary production and
remineralization within the estuary. This largely explains the discrepancy between watershed
model hypoxia estimates (Table 5).
Our findings demonstrate the importance of considering differences among these three
factors (ESM, downscaling, and watershed model) that may contribute to a wider range of target
water quality variables and living resource responses in coastal marine ecosystems like the
Chesapeake Bay that are highly influenced by watershed processes. Hydrological model
assumptions can have potentially significant impacts on estuarine hypoxia. For example, the
relatively high organic nitrogen loadings in Phase 6 compared to DLEM's comparatively modest
exports under the same future scenarios result in different levels of annual hypoxia. While
dramatic increases in organic nitrogen loadings within Bay tributaries are mostly limited to
Cool/Wet Phase 6 scenarios, there is precedent for catastrophic erosion within the Bay watershed
driven by extreme precipitation events (Springer et al., 2001). The relative uncertainty
introduced by individual factors is also not necessarily equivalent for streamflow, nitrogen
loadings, and AHV (Fig. 12). The complex connections between terrestrial runoff and
biogeochemical changes in the marine environment may expand further when higher order
trophic-level species are considered, and even more so when direct atmospheric impacts on the
Bay are also included. It is unlikely that general conclusions regarding the relative impacts of
different factors can be drawn for a marine ecosystem when only uncertainties in watershed
streamflow and nutrient loadings are considered. Had our results only accounted for the impacts
of these factors on watershed changes and not estuarine oxygen levels, the role of downscaling
could be incorrectly assumed to contribute negligible variability to hypoxic volume (Fig. 12). It
is the complex interactions of nitrogen species transformations within this estuarine model that
are responsible for this somewhat unexpected large contribution of downscaling method
uncertainty that is less prominent in watershed changes.
Despite the relatively small magnitude of Chesapeake Bay watershed climate impacts on
estuarine hypoxia compared to previous evaluations of other climate impacts, like atmospheric
warming over the Bay (Irby et al., 2018; Ni et al., 2019; Tian et al., 2021), the relative
contributions of ESM and downscaling effects to the total uncertainty are large and are also
likely to expand the range of outcomes for other climate sensitivity studies in this region. This
suggests that, when attempting to determine a likely range of ecosystem outcomes, selecting
additional downscaling techniques and hydrological model responses should be considered in
addition to the more common practice of only selecting multiple ESMs.
**4.2 Watershed Climate Scenario Impacts on Riverine Export and Hypoxia**
The climate scenario projections evaluated in this study are in near complete agreement that
the Chesapeake Bay watershed will be warmer and experience greater levels of precipitation by
mid-century, yet these results are not as straightforward to interpret as they relate to changes in
streamflow, nutrient loads, and estuarine hypoxia. Climate impacts on extreme river flows are
currently evident at global scales (Gudmundsson et al., 2021), and projected increases in
precipitation that could shape such events are aligned with estimates for this region derived from
observational (Yang et al., 2021) and modeling (Huang et al., 2021) studies, as well as for other
regions at similar latitudes (Bevacqua et al., 2021; Madakumbura et al., 2021). However,
differences exist in the spatial distribution and timing of these precipitation increases, as well as
in the temperature-affected rates of evapotranspiration. As a result, these estimates produce
varied projections for future freshwater streamflow. These complex interactions make it difficult
to directly predict future streamflow from projected precipitation changes, and even more
difficult to relate these to changes in nutrient loading. For example, in this study half of the
climate scenarios produce increasing streamflow on an annual basis, yet more than 75% of these
scenarios increase total nitrogen loading. Differences in the representation of soil and riverine
nitrogen processes between watershed models also results in inconsistent simulated responses of
nitrogen export to similar precipitation rates. Disparate export of nitrogen species (i.e., nitrate
and organic nitrogen) between watershed models also directly affects future nutrient load
projections. These hydrological model differences are evidenced by DLEM's higher $NO_3$ outputs
that offset lower organic nitrogen loadings (Fig. 7a).
Our analysis quantifies changes in hypoxia due to mid-century climate change impacts on the
watershed and provides an estimate of the relative uncertainty in these estimates. Our
experimental findings suggest that, in the absence of management actions, mid-century climate
impacts on the Chesapeake Bay watershed will increase hypoxia, specifically annual hypoxic
volume (AHV), by an average of $4 \pm 7\%$. This estimate is in good agreement with prior studies
that examined the impacts of watershed actions alone. Irby et al. (2018) applied a sensitivity
approach and projected increases in AHV of 5%, while Wang et al. (2017) showed increases in
annual anoxic volume of 9.7%, nearly equivalent to an increase of $10 \pm 16.5\%$ found here (Table
6). Results from this study also project that changes to Bay $O_2$ levels will vary spatially. Average
bottom main stem $O_2$ levels from May–September are expected to decrease most in the southern
half of the Bay (south of 38.5°N), particularly in climatologically dry years (Fig. 8).
Importantly, the projected changes presented here only account for the effects of climate
change on watershed response in isolation, and do not include the additional direct impacts of the
atmosphere and ocean. These additional changes have been estimated in other previous studies of
21[st] century impacts relative to observed conditions (Table 6). While numerous differing metrics
have been reported for many of these studies, including shifting dissolved oxygen concentrations
and water quality regulatory criteria, this work can be compared against previous results by
examining changes to annual hypoxic and anoxic volumes. The majority of these studies (Table
6) apply idealized changes to climate forcings and generally project increases in hypoxic
conditions. Increases in mid-21st century annual hypoxic volume due to watershed forcings
(+5% and $+4.4 \pm 7.4\%$) are smaller than average impacts of increasing temperatures alone

(+13%), while the results of changing sea level are more mixed (Table 6). However, the variability in hypoxia due to watershed changes is likely greatest among these factors and may substantially modify the negative effects of warming on dissolved oxygen concentrations. Our results and their uncertainties generally encompass the range of future hypoxia estimates found in previous research that have studied multiple climate impacts in isolation and in various combinations. Future work that accounts for the sources of uncertainty explored here by applying realistic climate change projections while also standardizing a metric for model results, like annual hypoxic volume, will help to narrow and better quantify definitive trends due to multiple factors that influence Bay dissolved oxygen.

Our findings are focused on Chesapeake Bay hypoxia, but some lessons can also be drawn from other coastal ecosystems where changes in watershed streamflow and nutrient loadings are also projected. In the Baltic Sea, Meier et al. (2011b) reported that hypoxia was very likely to increase regardless of ESM or climate scenario, assuming targeted reductions in accordance with the Baltic Sea Action Plan (decrease of nitrogen loads by $23 \pm 5\%$) were not met. Extensive studies of projected oxygen change in the Baltic Sea have repeatedly demonstrated that climate impacts are likely to increase hypoxic area (BACC II, 2015 and references therein), but more recent reports (Saraiva et al., 2019a; Wåhlström et al., 2020; Meier et al., 2021, 2022) have reaffirmed that nutrient reductions in accordance with the Baltic Sea Plan are also highly likely to mitigate a substantial amount of those hypoxia increases. Repeated investigations into the impact of increased streamflow and higher temperatures in the Gulf of Mexico demonstrate a likely expansion of hypoxic area (Justić et al. 1996; Lehrter et al., 2017; Laurent et al., 2018), and additional nutrient reductions required to mitigate these impacts (Justić et al., 2003). Finally, Whitney and Vlahos (2021) demonstrated a considerable erosion in oxygen gains in Long Island Sound due to nutrient reductions in the presence of climate effects, reducing projected mid-century improvements by 14%, similar to the 9% increase in hypoxic volume reported by Irby et al. (2018) for $O_2$ levels $< 2$ mg L$^{-1}$. Although these studies include direct climate change impacts on coastal water bodies, most support the findings here demonstrating that increases in streamflow and associated nutrient loadings are likely to increase Chesapeake Bay hypoxia. Overall, climate impacts on land have the potential to profoundly modify biogeochemical interactions in the coastal zone and limit the efficacy of nutrient reductions.

**4.3 Hypoxia Lessened by Impacts of Management Actions**

Projections of changes to watershed streamflow and nutrient delivery can better inform regional environmental managers tasked with managing interactions among nutrient reduction strategies, climate change, and coastal hypoxia (Hood et al., 2021; BACC II, 2015; Fennel and Laurent, 2018). The Chesapeake Bay results provided in this analysis demonstrate that the management actions mandated to improve water quality (USEPA, 2010) will decrease hypoxia by roughly 50%, approximately an order of magnitude more than projected increases due only to watershed climate change (Fig. 11). Therefore, nutrient reduction strategies are very likely to remain effective at reducing watershed nutrient loading and its contribution to eutrophication and hypoxia over a range of possible ESM scenarios (Mastrandrea et al., 2010). Should all management actions be implemented as outlined in the USEPA's Total Maximum Daily Load (USEPA, 2010), it is very likely that future climate impacts on Bay watershed runoff will worsen Bay hypoxia by a far smaller amount, relative to 1990s reference conditions. These findings are consistent with those of Irby et al. (2018) who also examined the impacts of watershed climate

on Chesapeake Bay hypoxia for the mid-21$^{st}$ century. When evaluating the effects of watershed climate impacts and management actions together, Irby et al. (2018) estimated an average AHV increase of 12.8 km$^3$ d, which is well within the range of 17.1 ± 34.8 km$^3$ d reported here (Table 6). Additionally, the combined impact of all climate stressors reported by Irby et al. (2018), i.e. atmosphere, ocean, and watershed, increased average AHV by 24.5 km$^3$ d, which is also within the range of the results reported here. Because climate change impacts are likely to increase total nitrogen loads, implementing nutrient reductions that do not account for the detrimental effects of climate change will reduce the likelihood of attaining water quality targets. Further quantifying a range of future estimates of watershed streamflow and nitrogen loading using regional models is critical to understanding the possibilities and limitations of mitigating negative climate impacts via nutrient reductions.

Recent findings support the hypothesis that nutrient reductions will improve water quality despite projected climate impacts in both freshwater systems (Wade et al., 2022) and other coastal marine systems (Whitney and Vlahos, 2021; Saraiva et al., 2019a; Bartosova et al., 2019; Wåhlström et al., 2020; Pihlainen et al., 2020; Meier et al., 2021; Große et al., 2020; Jarvis et al., 2022). In the Chesapeake Bay, reduced nutrient loading (Zhang et al., 2018; Murphy et al., 2022) has already helped mitigate growing climate change pressures (Frankel et al., 2022), despite rapidly increasing Bay temperatures over the past 30 years (Hinson et al., 2021). Like these prior studies, our findings confirm that management actions will likely produce even greater benefits to O$_2$ in coastal zones strongly affected by terrestrial runoff. While direct effects (e.g., air temperature) are expected to increase hypoxia more so than watershed changes in Chesapeake Bay (Irby et al., 2018, Ni et al., 2019), the comparatively greater impacts of management actions reported here are also likely to substantially reduce the overall risk from a multitude of co-occurring climatic stressors.

**4.4 Study Limitations and Future Research Directions**

Despite the plainly evident finding of nutrient reduction strategies improving water quality and counteracting negative climate change watershed impacts, a number of important caveats should temper this conclusion. First, the subset of scenarios that include management actions is limited to a set of five ESMs statistically downscaled with a single methodology and applied to one watershed model. As demonstrated in this work, this assumption may oversimplify the complex relationship between climate forcings and watershed model simulations, especially given that DLEM scenarios produce more change in nitrate and consequently more hypoxia than Phase 6 scenarios. Management actions implemented in Phase 6 nutrient reduction scenarios represent a multitude of possible methods to reduce point and nonpoint source pollution that are assumed to be fully implemented with a high operational efficacy by mid-century, but the true performance of best management practices operating under future hydroclimatic stressors remains largely unresolved (Hanson et al., 2022). Additionally, the importance of legacy nitrogen inputs to the Bay may grow over time (Ator and Denver, 2015; Chang et al., 2021), and can only be properly accounted for via a long-term transient simulation that accounts for changing groundwater conditions.

A key strength of the delta method applied here is its ability to remove the influence of interannual variability, which is known to strongly influence hypoxia in the Chesapeake Bay (Bever et al., 2013). However, the delta method is unable to account for the impacts of unanticipated extreme events, or changing patterns of precipitation intensity, duration, and

frequency that produce dramatic responses in sediment washoff, scour, and consequent
watershed organic nitrogen export. Air temperature and precipitation were the only watershed
model input variables adjusted in this analysis, allowing for a more equivalent comparison
between downscaling approaches. Future representations of watershed change may also better
account for changes in runoff through the inclusion of factors like ESM-estimated relative
humidity that can help avoid possible unreasonable amplification of potential evapotranspiration
that would decrease tributary streamflow (Milly and Dunne, 2011) and associated nutrient loads.
Although main stem Bay oxygen levels are the focus of this study, watershed impacts are
also likely to influence water quality in smaller scale tributaries. Differences in Chesapeake Bay
temperatures introduced by ESM and downscaling method have also been investigated by
Muhling et al. (2018) and contribute to biogeochemical variability via direct impacts of
atmospheric temperature on Bay warming. Incorporating different facets of these relative
uncertainties into projections of coastal change has also been demonstrated to affect ecological
outcomes like those surrounding fisheries (Reum et al., 2020; Bossier et al., 2021). Thus, the
impacts of these uncertainties are also very likely to affect socio-economic systems tied to
coastal resources. The analytical method applied here is well established within climatic and
terrestrial settings, so the relative dearth of coastal applications (excluding Meier et al., 2021)
may be more related to a consequence of computational demand or greater focus on uncertain
parameterizations of marine biogeochemical processes (Jarvis et al., 2022) that also play a large
role in potential future hypoxia outcomes.
**5 Conclusions**
Coastal ecosystems like the Chesapeake Bay that are currently and will likely continue to be
negatively affected by climate impacts exhibit complex responses in future scenarios,
demonstrating our lack of complete system understanding. While this research reaffirms the
importance of management actions in reducing levels of hypoxia, it also highlights the fact that
uncertainties in climate-impacted watershed conditions will affect estimates of Chesapeake Bay
$O_2$ levels. Additional study of uncertainty interactions within a full climate scenario (that
includes the impacts of changing atmospheric and oceanic conditions) will help better quantify a
range of hypoxia projections, among other environmental conditions within the Chesapeake Bay.
These results underscore the need for additional rigorous analyses of model parameterizations
and their contributions to model scenario uncertainty to help identify biogeochemical processes
that are most sensitive to climate change impacts and warrant further investigation. The
development of more rapid techniques to evaluate a broader range of future water quality and
ecological outcomes, and an inspection of their underlying assumptions, can help provide a
better mechanistic understanding of complex reactions to multiple climate stressors. Like
ongoing efforts to reduce greenhouse gas emissions and lessen the impacts of future climate
change globally, continuing efforts to reduce eutrophication in coastal waters could help improve
ecosystem resilience and the benefits derived by communities dependent on their function.
Nutrient reduction plans are likely to become even more essential to managers tasked with
preserving the health and function of rapidly evolving coastal environments and unfamiliar
future conditions.

**Appendix A:**

Original partitioning of organic nitrogen pools from the DLEM and Phase 6 watershed models was based on fixed fractions previously described in Frankel et al. (2022). There, 80% of the refractory organic nitrogen (rorN) loadings from Phase 6 were allocated to the small detritus nitrogen (SDeN) pool and the remainder was applied to the refractory dissolved organic nitrogen (rDON) pool in ChesROMS-ECB. More realistic changes to this partitioning of watershed rorN loadings were implemented, which decreased the lability of organic nitrogen loads overall. A specified threshold of rorN loadings was set at the 90th percentile of reference Phase 6 watershed inputs to the estuarine model, and thresholds were also set for individual river levels of streamflow at the 50th and 90th percentiles of Phase 6 reference simulations. Below the 50th percentile of streamflow levels, 80% of the rorN inputs below the specified rorN threshold were allocated to ChesROMS-ECB's SDeN pool, and the remainder were assigned to the rDON pool. Between the 50th and 90th percentiles of streamflow events, 50% of the rorN load below the specified rorN threshold was apportioned to ChesROMS-ECB's SDeN and rDON pools. At the uppermost levels of streamflow (greater than the 90th percentile), 5% of rorN was allocated to SDeN and 95% was given to rDON within ChesROMS-ECB. For any partitioning of an organic nitrogen load, regardless of the level of streamflow, rorN loading above this cutoff was allocated to ChesROMS-ECB's rDON pool. The rorN load below this threshold was allocated according to the fractionations described above. Changes to Phase 6 watershed loadings were mapped to equivalent DLEM watershed input variables, following the methodology of Frankel et al. (2022).

**Table A1. Acronyms and Abbreviations**

| | |
|---|---|
| AHV | annual hypoxic volume |
| BCSD | Bias-Correction and Spatial Disaggregation |
| CBP | Chesapeake Bay Program |
| ChesROMS-ECB | Chesapeake Regional Ocean Modeling System – Estuarine Carbon and Biogeochemistry |
| CMIP | Coupled Model Intercomparison Project |
| DIN | dissolved inorganic nitrogen |
| DLEM | Dynamic Land Ecosystem Model |
| DON | dissolved organic nitrogen |
| DSC | downscaling methodology |
| ESM | earth system model |
| KKZ | Katsavounidis-Kuo-Zhang (Katsavounidis et al., 1994) |
| MACA | Multivariate Adapted Constructed Analogs |
| Phase 6 | Phase 6 Watershed Model |
| RCP | representative concentration pathway |
| WSM | watershed model |

**Appendix B:**

An example calculation of the methodology used to calculate uncertainty for a single component of the total uncertainty is provided below. Average annual changes in hypoxic volume (km$^3$ d) are shown in the table below for the Multi-Factor experiment. Values of hypoxic volume are rounded to the tenth decimal place in Tables B1-B3, but the rounding is not carried through all calculations.

Table B1.

| ESM | P6 MACA | P6 BCSD | DLEM MACA | DLEM BCSD |
|------|---------|---------|-----------|-----------|
| KKZ1 | -34.3 | 34.6 | 53.4 | -2.0 |
| KKZ2 | -18.8 | 57.7 | 7.2 | -12.5 |
| KKZ3 | 24.8 | 23.8 | 139.2 | 71.8 |
| KKZ4 | -10.7 | -32.3 | 88.0 | 8.6 |
| KKZ5 | 64.7 | 93.7 | 24.3 | 94.3 |

For the first calculation, a subset of two ESMs is selected so that the number of values is balanced among ESMs, downscaling methods, and watershed models. This process will be repeated for each possible combination of ESMs, ten in total {(1,2), (1,3), (1,4), …, (4, 5)}.

Table B2.

| ESM | P6 MACA | P6 BCSD | DLEM MACA | DLEM BCSD |
|------|---------|---------|-----------|-----------|
| KKZ1 | -34.3 | 34.6 | 53.4 | -2.0 |
| KKZ2 | -18.8 | 57.7 | 7.2 | -12.5 |

For simplicity, the above table can be rearranged to that shown below. Additionally, the format of the table below and the following equations largely mirror the format of Ohn et al. (2021).

Table B3.

| Stage 1 (E) | Stage 2 (D) | Stage 3 (W) | $Y_x$ |
|-------------|-------------|-------------|-------|
| $x_{1,1}$ | $x_{2,1}$ | $x_{3,1}$ | -34.3 |
| | | $x_{3,2}$ | 53.4 |
| | $x_{2,2}$ | $x_{3,1}$ | 34.6 |
| | | $x_{3,2}$ | -2.0 |
| $x_{1,2}$ | $x_{2,1}$ | $x_{3,1}$ | -18.8 |
| | | $x_{3,2}$ | 7.2 |
| | $x_{2,2}$ | $x_{3,1}$ | 57.7 |
| | | $x_{3,2}$ | -12.5 |

First, the total variance of this subset ($U_{\{1,2,3\}}^{cumul}$) is calculated, with the subscripts of each individual factor (ESM=1, Downscaling Method=2, Watershed Model=3) denoted in brackets, and N defined as the total number of possible outcomes ($Y_x$ in Table B3):

$$U_{\{1,2,3\}}^{cumul} = \frac{1}{N}\sum_{i}^{N}(X_i - \bar{X})^2 = 1025.1$$

Following this, the cumulative uncertainty due to the choice of downscaling method and
watershed model ($U_{\{1,2\}}^{cumul}$) is calculated by selecting all $Y_x$ values from Table B3 where the first
two stages vary ($Y_{\{1,2\}}$) but the third stage does not (either ($x_{3,1}$) or ($x_{3,2}$)):

$$Y_{\{1,2\}}(x_{3,1}) = \{-34.3, 34.6, -18.8, 57.7\}$$
$$Y_{\{1,2\}}(x_{3,2}) = \{53.4, -2.0, 7.2, -12.5\}$$

$$U_{\{1,2\}}^{cumul} = \frac{1}{2}\left(U_{\{1,2\}}^{cumul}(x_{3,1}) + U_{\{1,2\}}^{cumul}(x_{3,1})\right) = \frac{1}{2}(1417.0 + 631.7) = 1024.3$$


Similar variance calculations are completed for the uncertainty of the first stage alone ($U_{\{1\}}^{cumul}$),
where the choice of ESM is the only constant:

$$Y_{\{1\}}(x_{2,1}, x_{3,1}) = \{-34.3, -18.8\}$$
$$Y_{\{1\}}(x_{2,1}, x_{3,2}) = \{53.4, 7.2\}$$
$$Y_{\{1\}}(x_{2,2}, x_{3,1}) = \{34.6, 57.7\}$$
$$Y_{\{1\}}(x_{2,2}, x_{3,2}) = \{-2.0, -12.5\}$$


Combining these values to calculate the uncertainty of the first stage alone (ESM) yields the
following equation, where i and j denote the factor choices from stages 2 and 3 in Table B3:
$$U_{\{1\}}^{cumul} = \frac{1}{4}\sum_{i=1}^{2}\sum_{j=1}^{2}\left(Y_{\{1\}}(x_{2,i}, x_{3,j})\right) = \frac{1}{4}(60.1 + 533.6 + 133.4 + 52.6) \approx 188.2$$


Applying similar calculations produces the following values necessary to compute total
uncertainty for all stages:

$$U_{\{1,2,3\}}^{cumul} = 1025.1$$
$$U_{\{1,2\}}^{cumul} = 1024.3$$
$$U_{\{2,3\}}^{cumul} = 1019.9$$
$$U_{\{1,3\}}^{cumul} = 947.7$$
$$U_{\{1\}}^{cumul} = 188.2$$
$$U_{\{2\}}^{cumul} = 877.7$$
$$U_{\{3\}}^{cumul} = 913.4$$


Next, the uncertainty of the first stage is calculated by subtracting the uncertainties from other
stages as follows:

$$U_{\{1,2,3\},1}^{cumul} = U_{\{1,2,3\}}^{cumul} - U_{\{2,3\}}^{cumul} = 5.2$$
$$U_{\{1,2\},1}^{cumul} = U_{\{1,2\}}^{cumul} - U_{\{2\}}^{cumul} = 146.6$$
$$U_{\{1,3\},1}^{cumul} = U_{\{1,3\}}^{cumul} - U_{\{3\}}^{cumul} = 34.3$$
$$U_{\{1\},1}^{cumul} = 188.2$$


The combined value of cumulative uncertainty for the first stage (ESM) can now be calculated:
$$\frac{1}{3}(U_{\{1,2,3\},1}^{cumul} + \frac{1}{2}U_{\{1,2\},1}^{cumul} + \frac{1}{2}U_{\{1,3\},1}^{cumul} + U_{\{1\},1}^{cumul}) = \frac{1}{3}(5.1 + 73.3 + 17.2 + 188.2) = 94.6$$


Applying the same computational steps results in cumulative uncertainties for stages 2
(Downscaling Method) and 3 (Watershed Model) of 475.5 and 480.5, respectively. These values
correspond to relative uncertainties for ESM, Downscaling Method, and Watershed Model of
9%, 45%, and 46%, respectively. This procedure is then repeated for all other combinations of
two ESMs {(1,3), (1,4), (1,5), …, (4, 5)}, after which the percentage values are averaged to
produce the estimates reported in our results.
**Competing Interests**: The authors declare that they have no conflict of interest.

**Author contribution**: MF, RN, HT, and GS were responsible for project conceptualization and
funding acquisition. MH, ZB, and GB were responsible for data curation used in the
experiments. KH and MF planned the model experiments; KH, MF, and PS are responsible for
the methodology (model creation). KH conducted the investigation and formal analysis and
created software and visualizations of results; KH wrote the original manuscript draft; MF, RN,
MH, ZB, GB, PS, HT, and GS reviewed and edited the manuscript.

**Acknowledgements**: This paper is the result of research funded by the National Oceanic and
Atmospheric Administration's National Centers for Coastal Ocean Science under award
NA16NOS4780207 to the Virginia Institute of Marine Science. Additional funding support was
provided by the VIMS Academic Studies Office. Feedback from principal investigators, team
members, and the Management Transition and Advisory Group of the Chesapeake Hypoxia
Analysis & Modeling Program (CHAMP) benefited this research. The authors acknowledge
William & Mary Research Computing for providing computational resources and/or technical
support that have contributed to the results reported within this paper
(https://www.wm.edu/it/rc). The authors also acknowledge the World Climate Research
Programme's Working Group on Coupled Modelling, which is responsible for CMIP, and we
thank the climate modeling groups for producing and making available their model output. For
CMIP, the U.S. Department of Energy's Program for Climate Model Diagnosis and
Intercomparison provides coordinating support and led development of software infrastructure in
partnership with the Global Organization for Earth System Science Portals. The model results
used in the manuscript are permanently archived at the W&M ScholarWorks data repository
associated with this article and are available for free download (https://doi.org/10.25773/5zet-
aq32). Finally, the authors would like to thank the anonymous reviewer and Bo
Gustafsson for their helpful and insightful comments that helped improve the manuscript.

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

**Tables and Figures**

**Table 1.** Experiments conducted to quantify future changes in Annual Hypoxic Volume (AHV).

| Experiment Name | Number of ESMs | Number of downscaling techniques | Number of watershed models | Number of simulations |
|---|---|---|---|---|
| Multi-Factor | 5[a] | 2 (MACA and BCSD) | 2 (DLEM and Phase 6) | 20[b] |
| Management | 5[a] | 1 (MACA) | 1 (Phase 6) | 5[c] |
| All-ESMs | 20 | 1 (MACA) | 1 (DLEM) | 20 |

[a]Corresponding to the KKZ-selected subset of five ESMs: Center, Cool/Dry, Hot/Wet, Cool/Wet, and Hot/Dry for both MACA
and BCSD downscaled model outputs.
[b]Additional scenarios were simulated for the Multi-Factor experiment as needed (for the Center and Hot/Wet ESMs) to
accurately partition uncertainty in model outcomes.
[c]An additional scenario simulated the effects of future management conditions without climate change impacts.
**Table 2:** Nash-Sutcliffe efficiencies of the DLEM and Phase 6 Watershed Models at the most
downstream stations of three major rivers, for monthly estimates of streamflow and nutrient
loading over the period 1991-2000. Nash-Sutcliffe efficiencies equal to one are indicative of
perfect model skill and negative values indicate that error variance exceeds the observed
variance.

| Major River | Freshwater Streamflow | | Nitrate Loading | | Organic Nitrogen Loading | |
|---|---|---|---|---|---|---|
| | DLEM | Phase 6 | DLEM | Phase 6 | DLEM | Phase 6 |
| Susquehanna | 0.74 | 0.88 | 0.60 | 0.78 | 0.37 | 0.12 |
| Potomac | 0.59 | 0.90 | 0.32 | 0.87 | 0.34 | -0.69 |
| James | 0.59 | 0.92 | -1.05 | 0.42 | 0.51 | 0.72 |


**Table 3:** Model skill metrics over the reference period (1991-2000)

| Variable | Depth | Watershed model | ChesROMS-ECB estimate | Observed estimate[a] | Bias | RMSD |
|---|---|---|---|---|---|---|
| $O_2$ [mg L$^{-1}$] | Surface | DLEM | $7.9 \pm 2.3$ | $9.3 \pm 2.0$ | -1.4 | 2.2 |
| | | Phase 6 | $8.0 \pm 2.3$ | | -1.4 | 2.2 |
| | Bottom | DLEM | $6.1 \pm 3.5$ | $5.7 \pm 3.5$ | 0.4 | 1.7 |
| | | Phase 6 | $6.2 \pm 3.4$ | | 0.5 | 1.6 |
| $NO_3$ [mmol N m$^3$] | Surface | DLEM | $0.32 \pm 0.36$ | $0.23 \pm 0.33$ | 0.09 | 0.23 |
| | | Phase 6 | $0.30 \pm 0.37$ | | 0.06 | 0.22 |
| | Bottom | DLEM | $0.27 \pm 0.33$ | $0.14 \pm 0.24$ | 0.13 | 0.25 |
| | | Phase 6 | $0.25 \pm 0.33$ | | 0.11 | 0.23 |
| DON [mmol N m$^3$] | Surface | DLEM | $0.27 \pm 0.05$ | $0.28 \pm 0.08$ | -0.00 | 0.08 |
| | | Phase 6 | $0.32 \pm 0.08$ | | 0.05 | 0.12 |
| | Bottom | DLEM | $0.27 \pm 0.05$ | $0.26 \pm 0.08$ | 0.00 | 0.08 |
| | | Phase 6 | $0.31 \pm 0.08$ | | 0.04 | 0.11 |
| Primary Production [mg C m$^{-2}$ d$^{-1}$] | Water Column | DLEM | $1146 \pm 154$[b] | $957 \pm 287$ | 189 | N/A |
| | | Phase 6 | $1133 \pm 129$ | | 176 | |
| AHV [km$^3$ d] | Water Column | DLEM | $987 \pm 254$ | $785 \pm 201$ | 202 | 250 |
| | | Phase 6 | $906 \pm 199$ | | 121 | 212 |

[a]Observed estimates and standard deviations for $O_2$, $NO_3$, and DON are from Water Quality Monitoring Program data
(Chesapeake Bay Program DataHub, 2022) at 20 main stem stations. Observed estimate and standard error for primary
production are derived from Harding et al. (2002), averaged over Feb-Nov for the years 1982-1998. Observed estimate and
standard deviation for AHV is derived by applying a weighted-distance interpolation model to observed $O_2$ at a limited number
of stations (Bever et al., 2013).
[b]Modeled primary production is computed only over Feb-Nov for comparison with the observed estimate.
**Table 4:** Annual average and standard deviations of reference (1991-2000) and climate scenario
(2046-2055) watershed loadings and estuarine hypoxia.

| Watershed Freshwater Streamflow [km$^3$ y$^{-1}$] | | | | | |
|---|---|---|---|---|---|
| Watershed Model | DLEM | | Phase 6 | | Phase 6 with Management |
| 1990s | 84 ± 26 | | 72 ± 21 | | 74 ± 21 |
| 2050s Downscaling | MACA | BCSD | MACA | BCSD | MACA |
| Center | 87 ± 28 | 74 ± 24 | 78 ± 21 | 80 ± 22 | 79 ± 21 |
| Cool/Dry | 76 ± 24 | 75 ± 24 | 67 ± 19 | 77 ± 22 | 68 ± 19 |
| Hot/Wet | 84 ± 29 | 86 ± 29 | 79 ± 22 | 77 ± 21 | 80 ± 22 |
| Hot/Dry | 77 ± 25 | 74 ± 23 | 70 ± 20 | 68 ± 20 | 72 ± 20 |
| Cool/Wet | 93 ± 29 | 95 ± 30 | 83 ± 22 | 80 ± 22 | 84 ± 22 |
| ESM Average | 84 ± 27 | 81 ± 26 | 75 ± 21 | 76 ± 21 | 77 ± 21 |
| Watershed Nitrogen Loading [10$^9$ gN y$^{-1}$] | | | | | |
| Watershed Model | DLEM | | Phase 6 | | Phase 6 with Management |
| 1990s | 151 ± 49 | | 147 ± 46 | | 87 ± 28 |
| 2050s Downscaling | MACA | BCSD | MACA | BCSD | MACA |
| Center | 159 ± 46 | 138 ± 41 | 177 ± 63 | 192 ± 75 | 103 ± 36 |
| Cool/Dry | 137 ± 39 | 132 ± 38 | 133 ± 36 | 166 ± 53 | 78 ± 23 |
| Hot/Wet | 157 ± 48 | 153 ± 45 | 183 ± 66 | 184 ± 68 | 105 ± 37 |
| Hot/Dry | 149 ± 45 | 138 ± 41 | 146 ± 42 | 140 ± 40 | 86 ± 26 |
| Cool/Wet | 159 ± 43 | 181 ± 62 | 301 ± 186 | 352 ± 244 | 156 ± 85 |
| ESM Average | 152 ± 43 | 148 ± 48 | 188 ± 110 | 207 ± 139 | 105 ± 53 |
| Annual Hypoxic Volume [km$^3$ d] | | | | | |
| Watershed Model | DLEM | | Phase 6 | | Phase 6 with Management |
| 1990s | 987 ± 254 | | 904 ± 171 | | 449 ± 144 |
| 2050s Downscaling | MACA | BCSD | MACA | BCSD | MACA |
| Center | 1072 ± 233 | 985 ± 250 | 926 ± 152 | 938 ± 152 | 470 ± 131 |
| Cool/Dry | 994 ± 252 | 975 ± 257 | 885 ± 177 | 961 ± 170 | 429 ± 148 |
| Hot/Wet | 1094 ± 247 | 1059 ± 249 | 931 ± 156 | 928 ± 171 | 480 ± 131 |
| Hot/Dry | 1075 ± 263 | 996 ± 259 | 893 ± 164 | 871 ± 165 | 442 ± 141 |
| Cool/Wet | 1011 ± 204 | 1081 ± 238 | 969 ± 170 | 997 ± 203 | 507 ± 138 |
| ESM Average | 1049 ± 234 | 1019 ± 244 | 921 ± 160 | 939 ± 171 | 466 ± 135 |


**Table 5:** Average ± standard error in ΔAHV (%) holding scenario effects (ESM, Downscaling
Method, Watershed Model) constant.

| Scenario Factor | Effect | Δ AHV % |
|---|---|---|
| ESM | Center | 4.4 ± 5.4 |
| | Cool/Dry | 0.9 ± 4.3 |
| | Hot/Wet | 6.7 ± 6.2 |
| | Hot/Dry | 1.4 ± 3.6 |
| | Cool/Wet | 8.3 ± 6.5 |
| Downscaling | MACA | 4.8 ± 6.0 |
| | BCSD | 3.9 ± 5.9 |
| Watershed Model | DLEM | 5.6 ± 7.5 |
| | Phase 6 | 3.1 ± 3.8 |


**Table 6:** A summary comparison of simulated mid-21st century climate change impacts on Chesapeake Bay hypoxia relative to observed conditions.

| Published Research | Climate Change Factors | Future Oxygen Change |
|---|---|---|
| **Watershed Changes** | | |
| Wang et al., 2017 | Increased watershed nitrogen loadings by +5 to +10% | No AHV estimate provided<br>Increase in AAV*: +9.7 to +18.7% |
| Irby et al., 2018 | Changed watershed streamflow by -2% to +17% (varying by month); assumed nutrient reductions | Increase in AHV: +5% |
| Hinson et al., 2023** (this paper) | Changed watershed streamflow and loadings according to two watershed models, two downscaling techniques, and five ESMs | Increase in AHV: +4.4 ± 7.4%<br>Increase in AAV: +10.0 ± 16.5% |
| **Temperature Changes** | | |
| Irby et al., 2018 | Increased estuarine temperatures by 1.75 °C; assumed nutrient reductions | Increase in AHV: +13% |
| Tian et al., 2021 | Increased atmosphere and ocean temperature increased by ~1 °C | †Increase in AHV: +9% |
| **Sea Level Rise** | | |
| Irby et al., 2018 | Increased sea level by 0.5 m; assumed nutrient reductions | Decrease in AHV: –13% |
| St-Laurent et al. 2019 | Increased sea level by 0.5 m for 4 different models | Increase in summertime bottom $O_2$ in all 4 models |
| Cai et al., 2022 | Increased sea level by 0.5 m | Increase in AHV by +8% |
| Cerco and Tian, 2022 | Increased sea level by 0.22 to 1 m and simulated wetland losses | Increase in DO criteria exceedances |
| **Multiple Environmental Changes** | | |
| Irby et al. 2018 | Combined atmosphere, watershed and sea level change, assuming nutrient reductions | Increase in AHV: +9% |
| Ni et al., 2019** | Combined atmosphere, watershed, and ocean Change: Multiple downscaled scenarios that increased air temperatures, monthly streamflow, ocean temperatures and sea surface height | Increase in AHV: +9 to 31%<br>Increase in AAV: +2 to 29% |
| Basenback et al., 2022 | Modified timing of nutrient delivery and warming within the estuary | Change in AHV: -10% to +18% |

AAV = Annual Anoxic Volume; AHV = Annual Hypoxic Volume

*AAV defined as $O_2 < 1$ mg $L^{-1}$ in Wang et al. (2017), and $O_2 < 0.2$ mg $L^{-1}$ for all others.

**Applied downscaled ESMs in projecting changes to Chesapeake Bay hypoxia.

†No 2050 estimate provided; results based on 2025 projected changes.

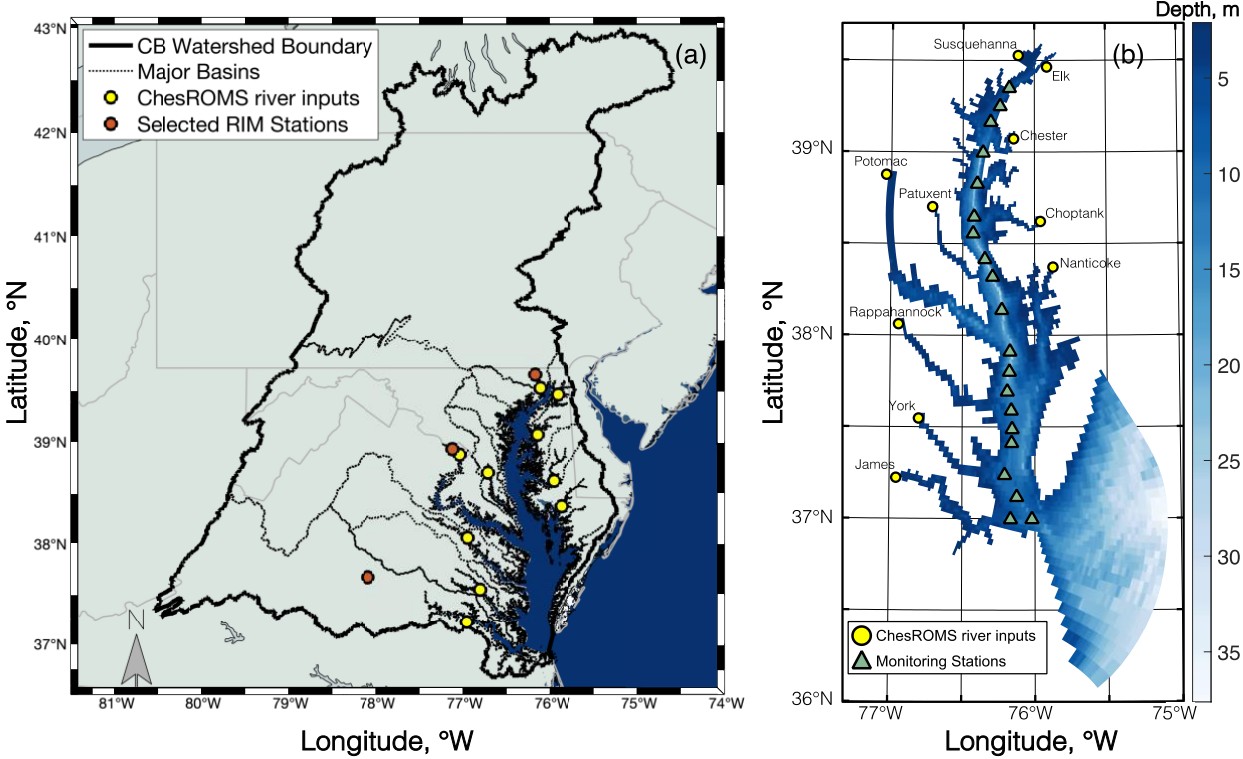

**Figure 1:** (a) Map showing the extent of the Chesapeake Bay watershed boundary, major basins,
River Input Monitoring (RIM) stations for the Susquehanna, Potomac, and James Rivers (red
circles), and ChesROMS-ECB river input locations (yellow circles). (b) Bathymetry of the
ChesROMS-ECB model domain, river input locations (yellow circles), and 20 Chesapeake Bay
Program main stem monitoring stations (green triangles). Base map layers from Pawlowicz
1334 (2020).

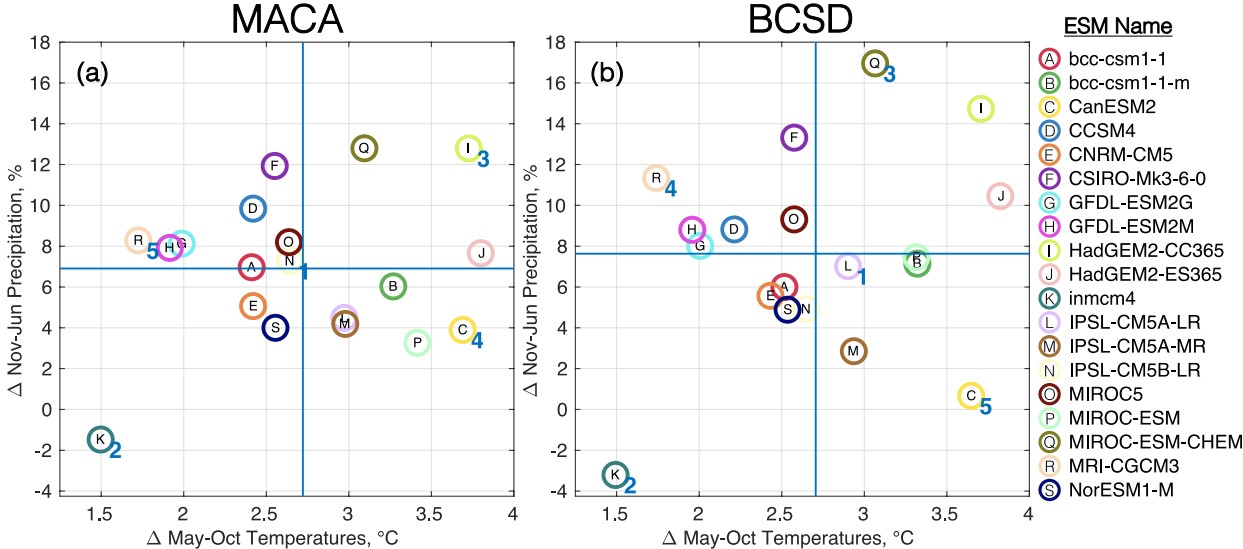

**Figure 2:** Relative changes in May-October temperatures and November-June precipitation over the Chesapeake Bay watershed for an ensemble of ESMs (circled letters) downscaled using (a) MACA and (b) BCSD methodologies. Horizontal and vertical blue lines correspond to the ensemble average changes in temperature and precipitation. Numbers adjacent to particular ESMs in both panels denote the order in which the first five were selected by the KKZ algorithm.

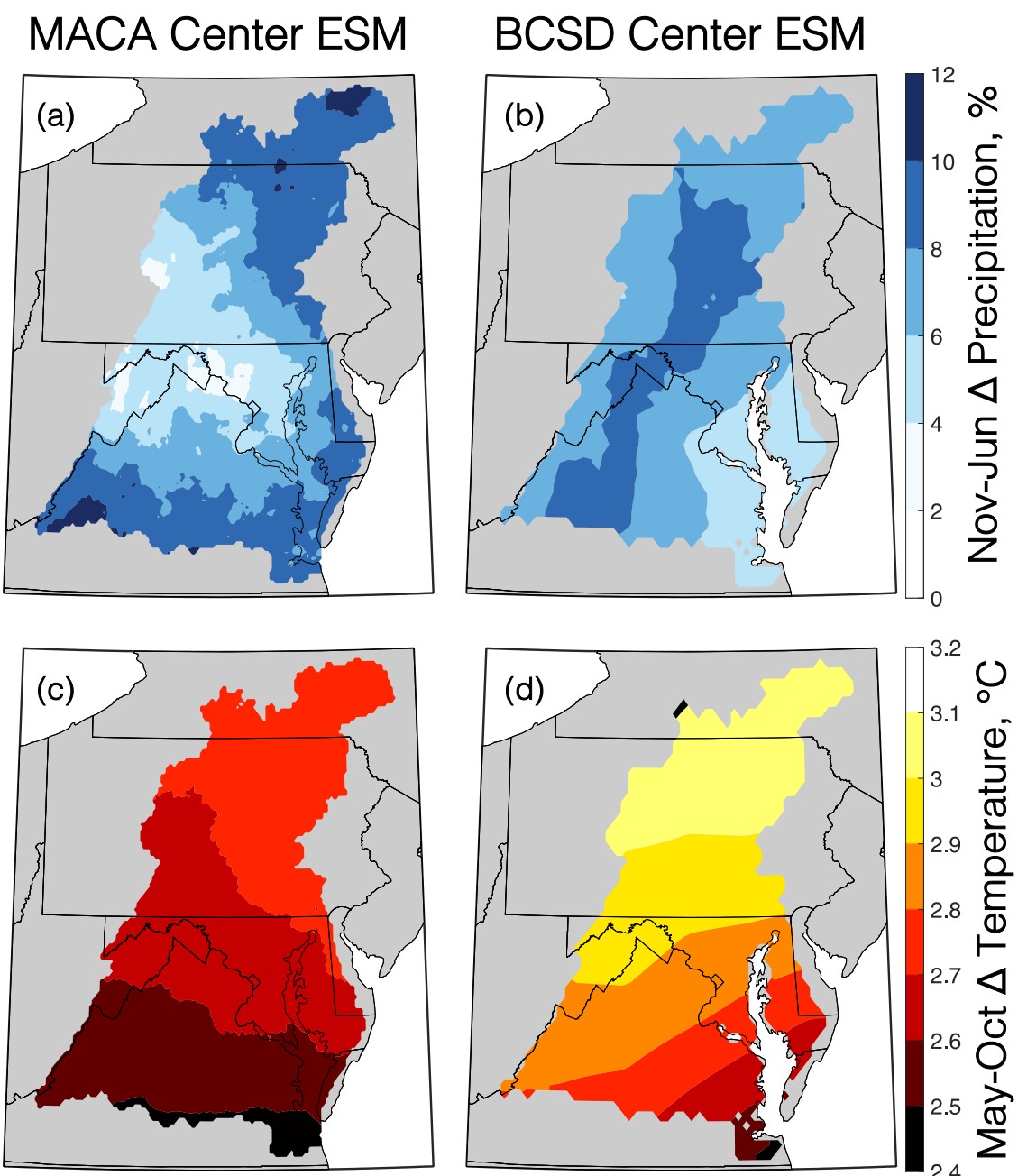

**Figure 3:** Changes in November to June precipitation (a, b) and May to October temperatures (c,
d) for the MACA (a, c) and BCSD (b, d) Center ESMs between mid-century (2046-2055) and the
reference period (1991-2000). Base map layers from Pawlowicz (2020).

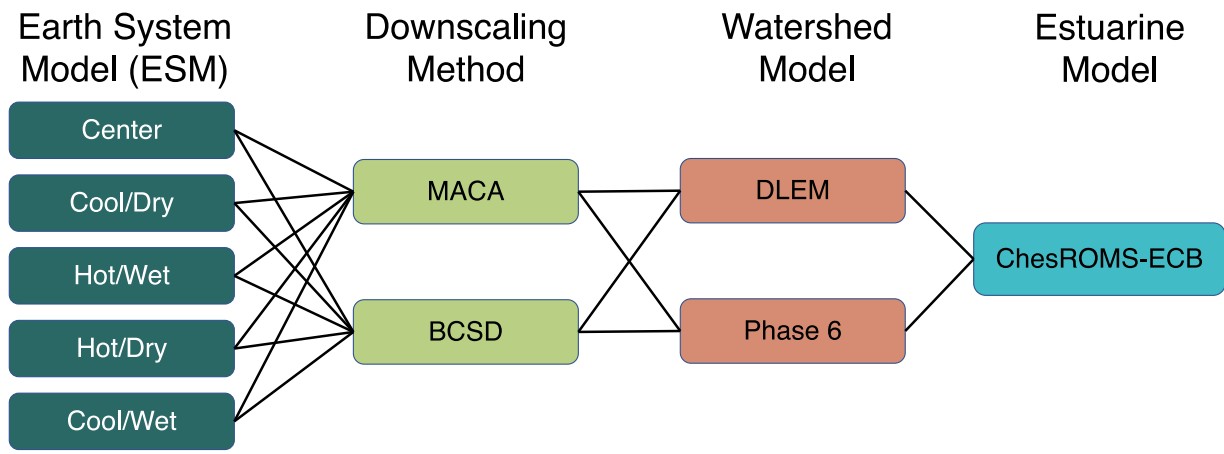

**Figure 4:** Diagram of Multi-Factor experimental design, comprising a total of 20 model
scenarios.

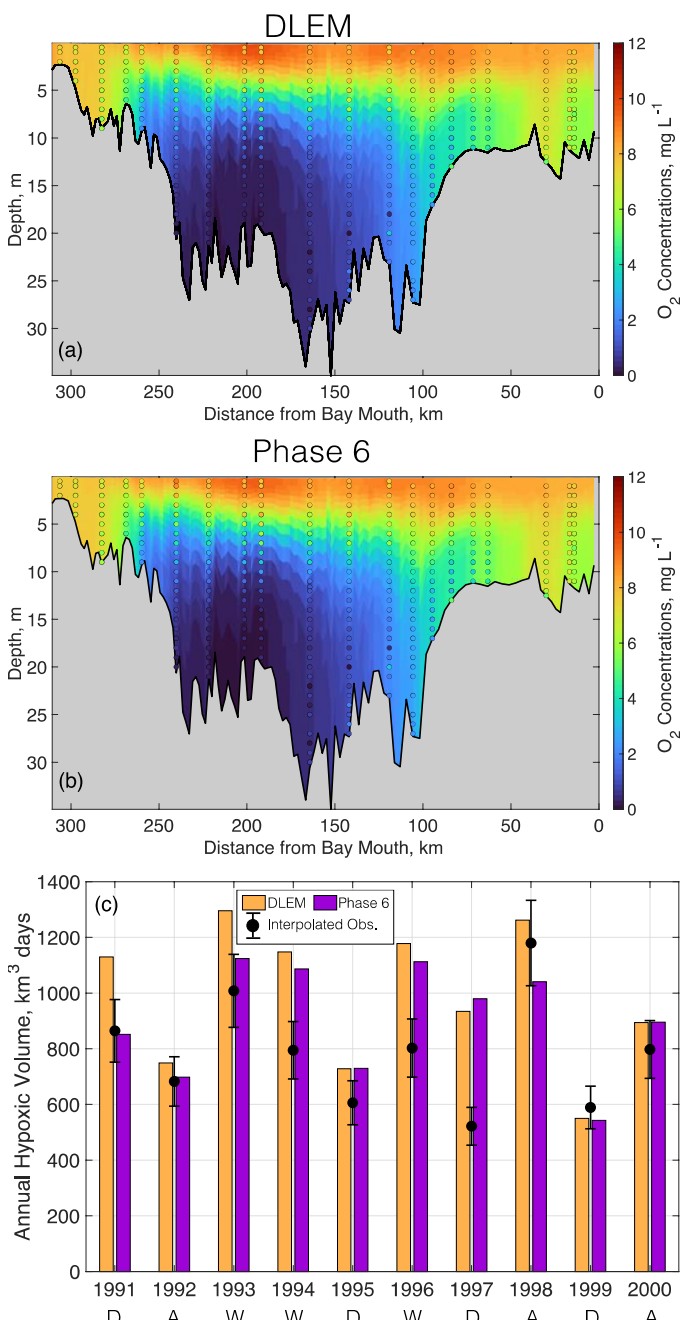

**Figure 5:** ChesROMS-ECB skill for average summer (Jun-Aug) $O_2$ profiles at main stem
monitoring locations using watershed inputs from (a) DLEM and (b) Phase 6 over the reference
period 1991-2000. (c) Modeled AHV using DLEM and Phase 6 compared to interpolated
observations (error bars denote RMS error) over the reference period 1991-2000. Average
hydrologic conditions are noted below corresponding years and signify dry (D), average (A), or
wet (W) years.

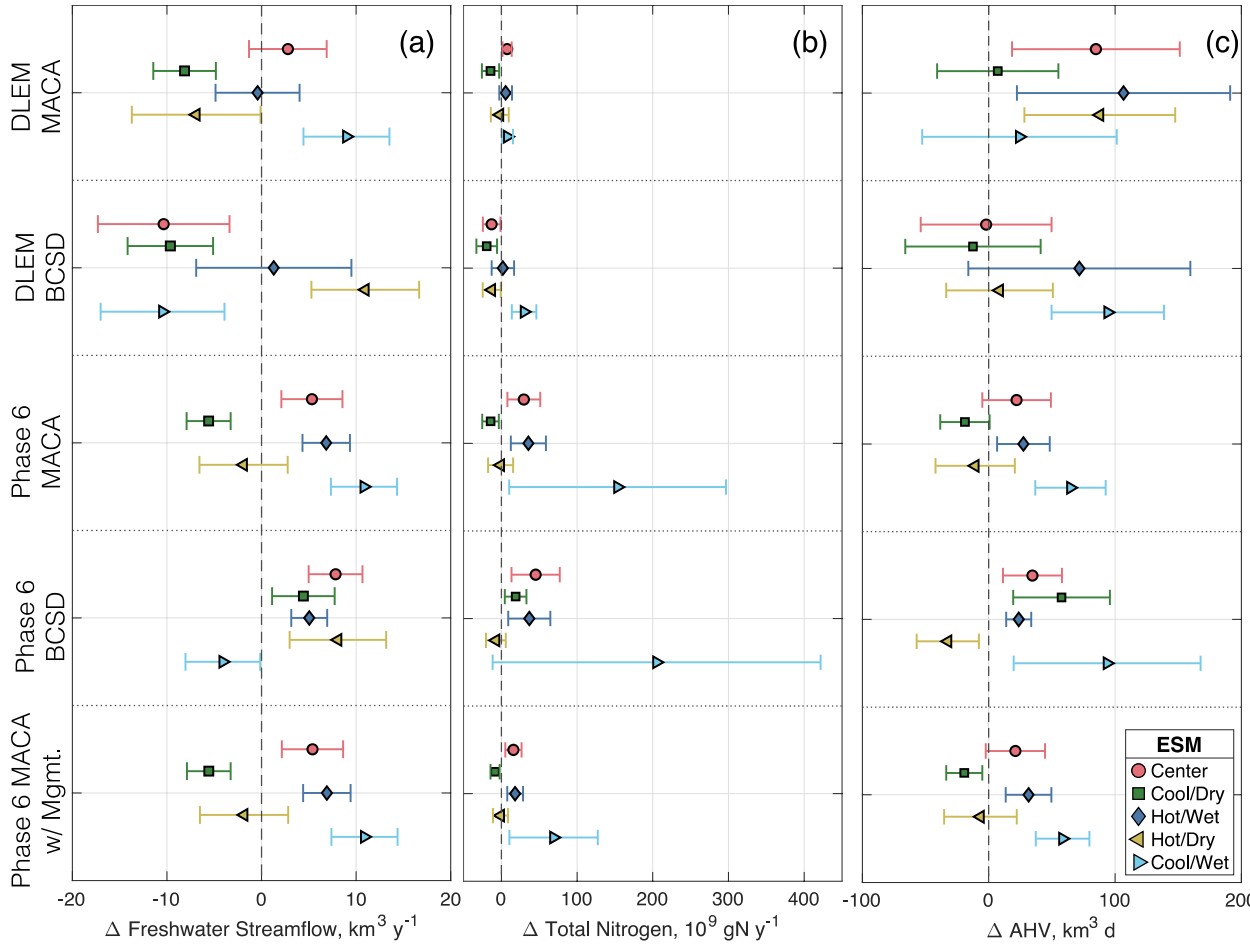

**Figure 6:** Mean and standard deviations of changes to freshwater streamflow (a), total nitrogen loadings (b), and annual hypoxic volume (c) for Multi-Factor and Management experiments. Future climate changes in these outputs are shown relative to 1990s baseline conditions (dashed vertical line) without management actions (upper four rows) and with management actions (bottom row).

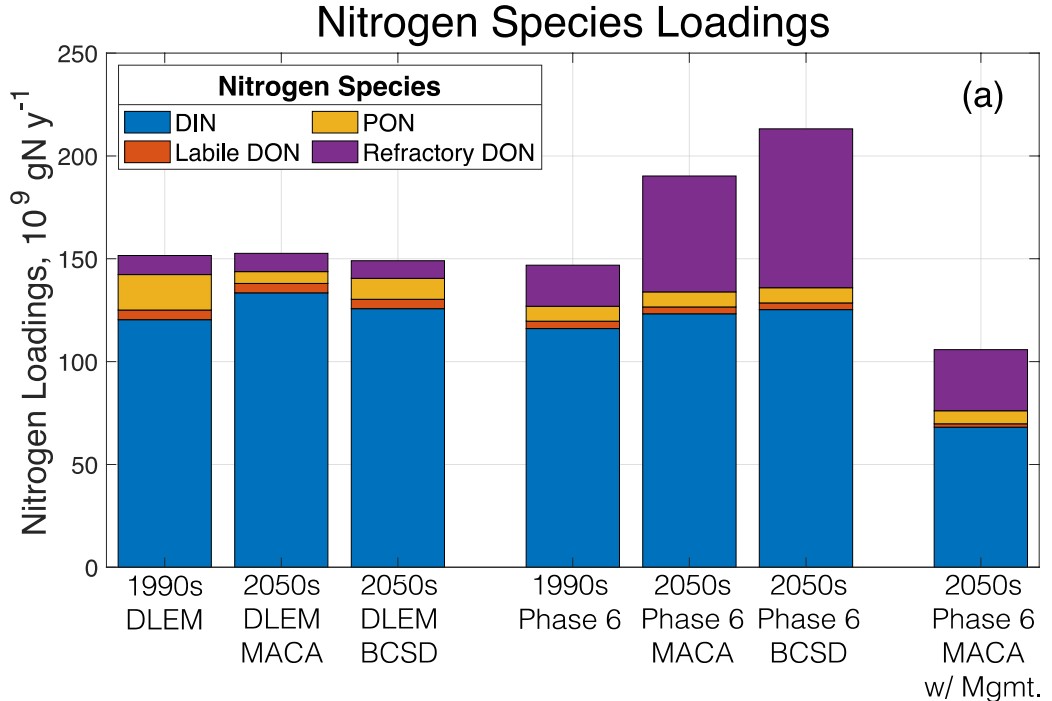

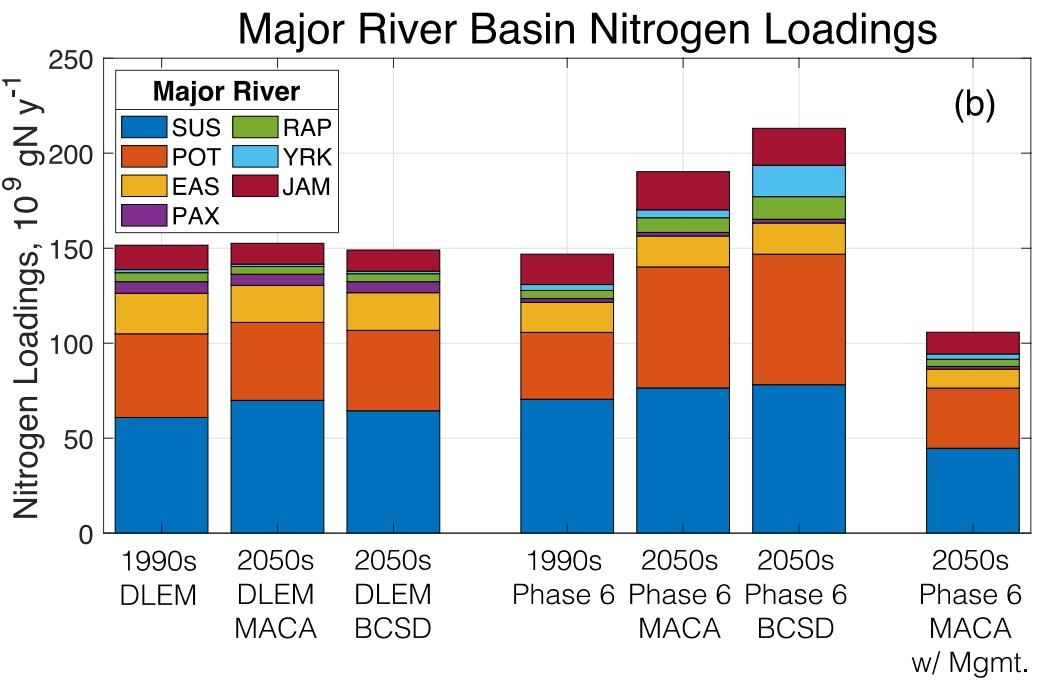

**Figure 7:** Average total nitrogen loadings among ESM scenarios for reference scenarios and various components of the Multi-Factor and Management experiments. Total nitrogen loadings divided by (a) nitrogen species component: dissolved inorganic nitrogen (DIN), particulate organic nitrogen (PON), dissolved organic nitrogen (DON), and refractory dissolved organic nitrogen, and (b) by major river basin (SUS = Susquehanna, RAP = Rappahannock, POT = Potomac, YRK = York, EAS denoting eastern shore rivers including the Elk, Chester, Choptank, and Nanticoke, JAM = James, PAX = Patuxent).

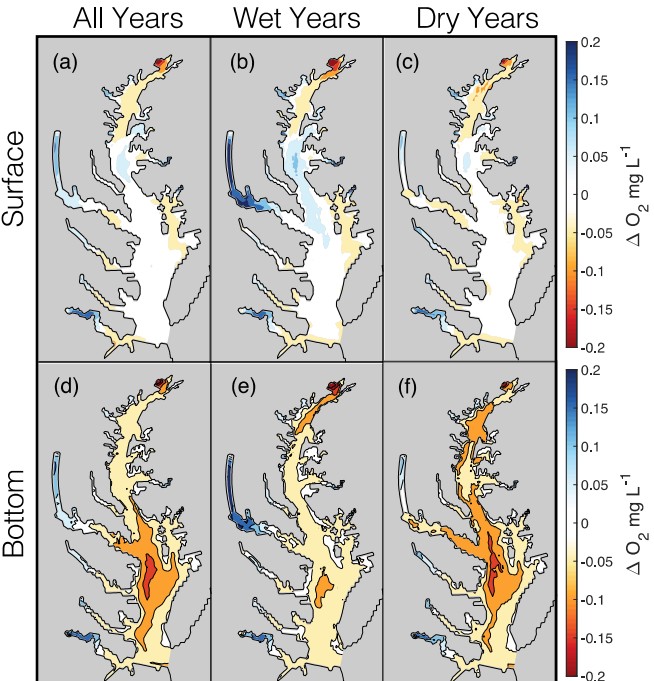

**Figure 8:** Average $O_2$ changes in Multi-Factor experiment scenarios at the surface (a-c) and
bottom (d-f) of the water column. Columns correspond to average changes for all years (a, d) as
well as hydrologically wet (b, e) and dry (c, f) years.

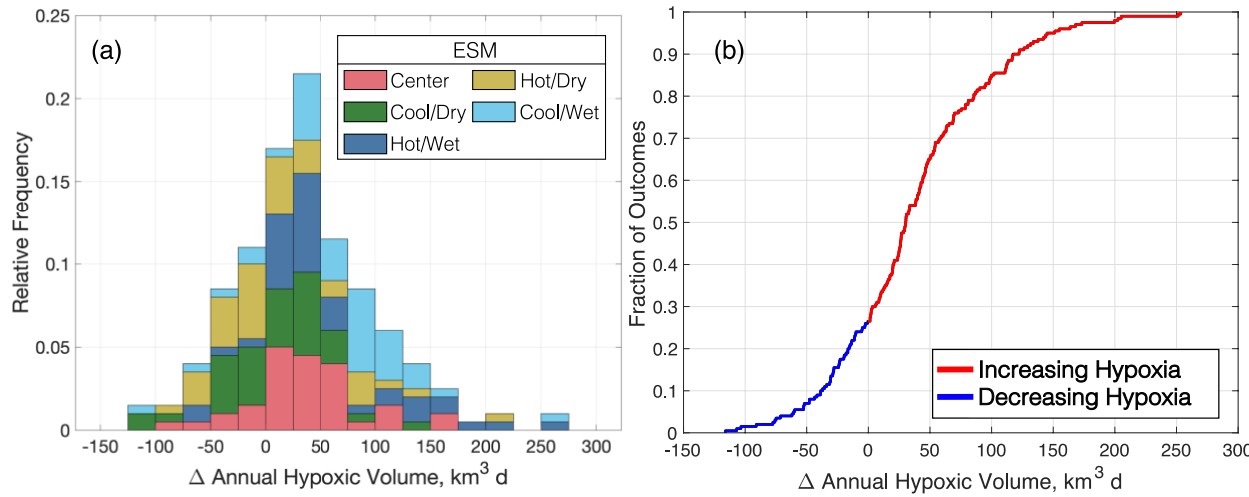

**Figure 9:** Summary of Multi-Factor experiment results for changes to Annual Hypoxic Volume,
expressed as a histogram of relative frequencies (a) and an empirical cumulative distribution (b).

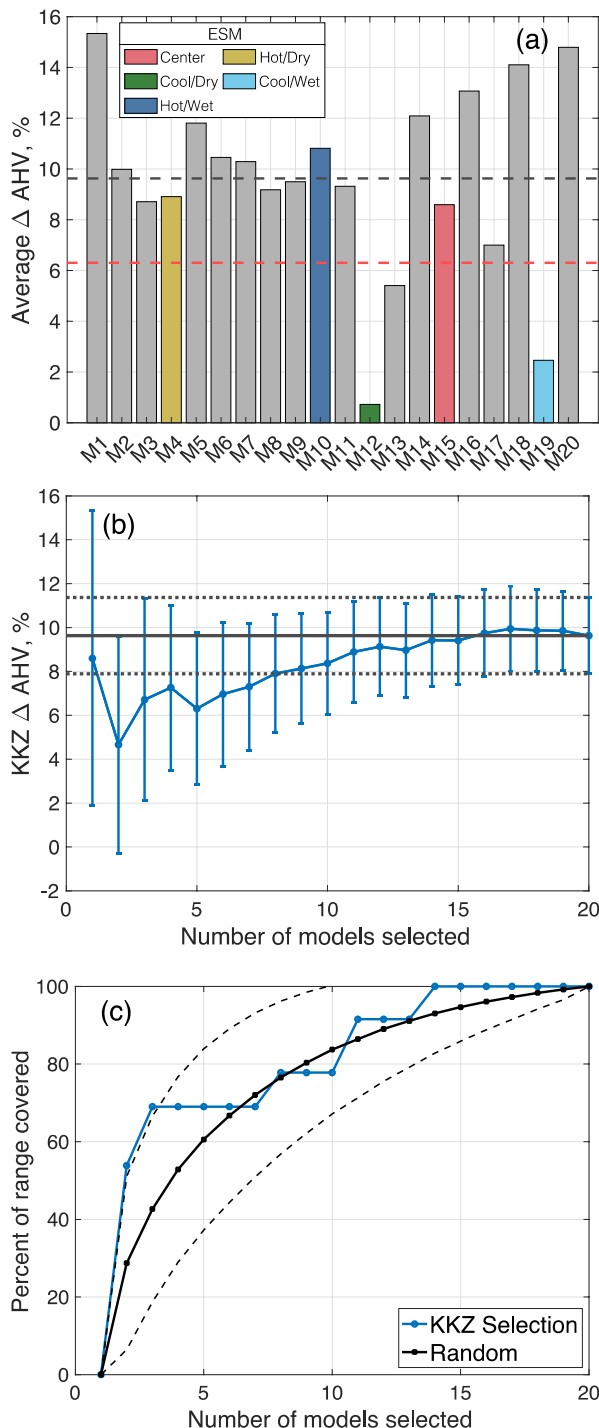

**Figure 10:** (a) Change in Annual Hypoxic Volume (ΔAHV) for the All-ESMs experiment. Red dashed line denotes the multi-model average of five KKZ-selected ESMs; black dashed line denotes the full 20-model average. (b) ΔAHV and standard errors as estimated by increasing number of KKZ-selected ESMs. Black lines correspond to 20-model average (solid) and associated standard errors (dotted) from the All-ESMs experiment. (c) Percent of ΔAHV range covered by sequentially increasing the number of KKZ-selected ESMs. Black lines correspond to the range (solid) and associated standard error (dashed) of estimates chosen by randomly selecting ESMs.

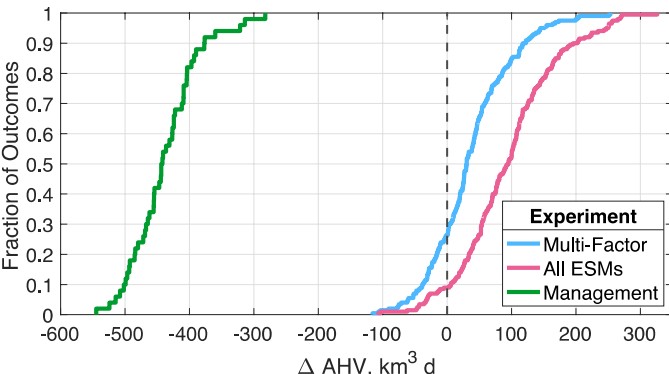

**Figure 11:** Summary of all experiment results for change in Annual Hypoxic Volume (ΔAHV), expressed as a cumulative distribution function. Black dashed vertical line corresponds to no change in AHV.

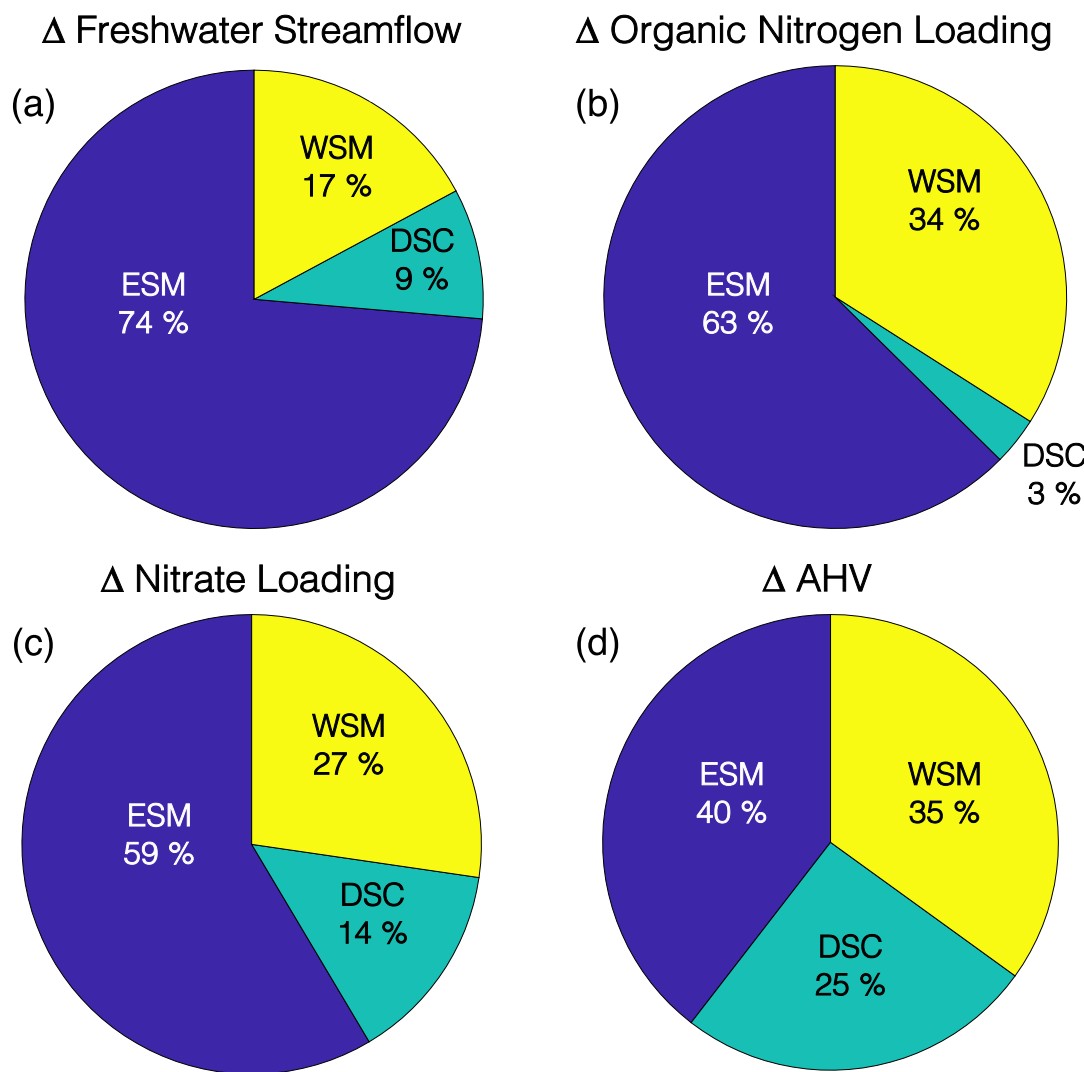

**Figure 12:** Percent contribution to uncertainty from Earth System Model (ESM), downscaling methodology (DSC), and watershed model (WSM) for estimates of (a) freshwater streamflow, (b) organic nitrogen loading, (c) nitrate loading, and (d) change in annual hypoxic volume (ΔAHV).