# Peer review of "Impacts and uncertainties of climate-induced changes in watershed inputs on estuarine hypoxia"

_EGUsphere, 2022_

## Author Response (AR2)

Biogeosciences Response to Reviewers

**Reviewer #1 Comments:**
In this study, the uncertainties in projections for river discharge, nutrient inputs and hypoxic volume of Chesapeake Bay have been studied in detail. The presented approach utilizes 20 Earth System Model (ESM) simulations of CMIP5, two statistical downscaling methods, two watershed models and a coupled physical-biogeochemical estuarine model for Chesapeake Bay. Projections for the mid-century under RCP8.5 and their uncertainties caused by differences in ESM, downscaling method and watershed model have been studied. The direct impact of climate change on the physics and biogeochemical processes in Chesapeake Bay were not considered. The authors found that all three factors significantly contribute to the uncertainty associated with future hypoxia, with the choice of ESM being the largest contributor. Furthermore, it was concluded that management actions have a bigger impact than climate change on river discharge and nutrient loads with the latter counteracting nutrient load abatement strategies.

- We would like to thank the reviewer for these helpful and positive comments, which will help clarify and strengthen our manuscript! Relevant comments and suggestions from the reviewer are pasted below, followed by a bullet summarizing our responses.

The study is very interesting and the results compare well with findings from other coastal seas with similar environmental conditions. The reference list is comprehensive and the strategy to reduce the number of ensemble members with the help of the KKZ algorithm applied to changes in air temperature and precipitation is very promising. The text is well written and the figures are of good quality.

- We thank the reviewer for these positive comments!

I only think that the large number of abbreviations makes the reading a little bit difficult. I suggest to add a table with abbreviation explanations. Furthermore, I suggest to also explain the abbreviations in the figure captions to enable an independent consideration of figures and text.

- The authors agree. A table defining our abbreviations will be added, and abbreviations defined in captions as the reviewer suggests. Some of the abbreviations can be removed or reiterated where necessary, e.g., TN for total nitrogen in Figure 7 and WQMP can be spelled out as it is only used twice.

It would also be nice to have a table summarizing all available projections of hypoxic volume of Chesapeake Bay from this study and the literature including columns for models, downscaling technique, emission and nutrient input scenarios, and changes in hypoxic volume and their uncertainties. However, the latter is just a suggestion.

- The authors considered this possibility, but the results are very difficult to compare comprehensively. Many previous Chesapeake Bay publications do not use a hypoxic volume (or any other) common metric.

I recommend to accept this nice manuscript for publication with minor revisions as outlined above. Future work should address a systematical comparison of projections of various coastal seas suffering from oxygen depletion.

- Thank you, and we agree about the direction of future work, including the development of a synthesis publication!

**Reviewer #2 Comments:**

This study aims to constrain the impact on and uncertainty of nutrient inputs, and subsequent consequence on hypoxia in the Chesapeake Bay from "pure" climate change in the catchment. The method is ensemble modeling, using outputs from several ESMs, different downscaling methods, and two different catchment models. All three steps in this chain comes with significant uncertainty, and the propagation of these uncertainties are quantified in terms of changes of the hypoxia in the Bay. The important overall conclusion with significant management implications is that climatic change impact on the catchment will not contribute substantially to increased hypoxia in the bay while that implementation active measures to reduce the inputs in the catchment will still provide significant improvements.

- We would like to thank Dr. Gustafsson for these helpful and positive comments, which will help clarify and strengthen our manuscript! Relevant comments and suggestions from the reviewer are pasted below, followed by a bullet summarizing our responses.

The paper is well written and I am especially impressed by the clear presentation of the massive amounts of simulations and analysis of these in the result section. At first, I was a somewhat sceptic to include the marine model and effects on hypoxia, but I have changed my opinion and agree now that it provides added value to the paper.

- Thank you very much for these positive comments!

The analysis is sound and statistics adequate. Models and methods are appropriately described, with some minor exceptions listed below.

- We very much appreciate the reviewer's minor comments, and we describe how these will be addressed below.

The discussion highlights the main key points, although I think you could consider to rearrange 4.1 and 4.2 so that the implications your main results are reiterated somewhat more.

Now the reader is not helped that much with interpretation of current results before other studies/regions are discussed. I am thinking of a focused discussion of the implications from primarily the complexity of the AHV response to scenarios (Fig 6) that is dependent on multitude of factors, among those the differences in the nitrogen load composition (Fig 7) leading ultimately to the very interesting uncertainty distribution in Fig 12.

- We agree with this suggestion and will switch the order of sections 4.1 and 4.2. We feel this helps to better emphasize the implications of our main results, as the reviewer suggests.

Below follows some minor comments:

*Introduction:*

A very comprehensive review of impact on hypoxia from combined climate/eutrophication scenarios, however, as I think the paper predominantly are about the catchment, I think it is adequate also to discuss Bartosova et al, 2019 and Pihlainen et al, 2020.

- We will add those citations to the relevant sections in the introduction and discussion.

*Methods:*

**246-274:** It is clever to use the KKZ selection method to reduce the number of simulations while ensure a wide spread in the results. However, I became quite interested in the method and read the original paper by Katsavounidis et al. with considerable difficulty, and thereafter Ross & Najjar that not were very much more helpful. The concept when you only have two variables is extremely simple and illustrated well in your figure 2, so I think you help the reader that is not so interested in exactly how it is computed by introducing figure 2 directly, and thereafter describe how these points were found.

- We will rearrange the order of this paragraph to first describe the figure and then refer to the method and its previous applications.

**280:** just to be sure: The only changes to the forcing of the ChesROMS-ECB scenarios are nitrogen and sediment loads from land and river runoff from land?

- This is exactly correct.

**292- 301:** It is somewhat unclear how the "second" and "third" experiments are implemented. Are nutrient inputs kept at TMDL and runoff varies or are different management actions implemented in the catchment, or???

- The second experiment, focused on management impacts, first simulates a baseline scenario that incorporates TMDL nutrient reductions without climate change effects using the Phase 6 Watershed Model. A set of five future climate experiments (downscaled with MACA and applied to the Phase 6 Watershed Model) that also include TMDL nutrient reductions are then compared to this reduced nutrient baseline scenario. In the third experiment, all 20 MACA-downscaled ESMs are applied to the DLEM watershed model to simulate future climate and better compare the effect of subsampling only 5 ESMs in the first experiment. In this third experiment, nutrient reductions are also not included. We will clarify where the TMDL nutrients are held constant and where they are not.

**304-322:** I find the description of what you do quite difficult to understand. Perhaps I am somewhat ignorant, and that it would be totally clear if I read the referred papers. However, I would appreciate a simpler explanation.

- The methodology is primarily based around calculating the relative variance due to ESM, downscaling method, and watershed model that is outlined in Bosshard et al. (2013). However, Ohn et al. (2021) provide a technique that can account for the undetermined interaction term from Bosshard. We agree that the methodology is a little complex, but we will add a brief section to the appendix demonstrating the steps of an example calculation used in the paper.

*Results*

**Fig 6 caption:** it would be good with a few more explanatory words in this caption. It took me a little while and comparison with Table 4 to understand that these results are deviations from the 1990s simulation for each catchment model/downscaling combination.

- A more detailed description will be added to the figure caption to ease interpretation.

**447-457:** I find it somewhat difficult to understand the point of the linear regression. However, I think that it may be partly because I do not really understand how the management scenarios are computed in the catchment model and if non-linear responses could have been expected.

- The point of the linear regression was to show that the same relative increases in future hypoxia occur for Phase 6 scenarios with/without nutrient reductions. Phase 6 Watershed Model experiments that include TMDL nutrient reductions apply a lower amount of nutrients to the terrestrial environment, but the relative magnitude of export to the estuary appears to remain the same. This finding highlights that current assumptions of nutrient reductions may need to be re-examined (with regards to nutrient reduction efficacy of best management practices) and the incorporation of more dynamic terrestrial processes (like those found in DLEM) may also need to be considered. This will be clarified further in the text.

Bartosova, A., R. Capell, J. E. Olesen, and others. 2019. Future socioeconomic conditions may have a larger impact than climate change on nutrient loads to the Baltic Sea. AMBIO: A Journal of the Human Environment **48**: 1325–1336. doi:10.1007/s13280-019-01243-5

Pihlainen, S., M. Zandersen, K. Hyytiäinen, and others. 2020. Impacts of changing society and climate on nutrient loading to the Baltic Sea. Science of the Total Environment **731**: 138935. doi:10.1016/j.scitotenv.2020.138935

**Editor Comments**

Both reviews of the original submission and my reading of the original submission were all quite positive. I determined that the revisions sufficiently address the comments of the reviewers with one exception. I think more can be done to address the reviewer comment about how do these results reported here relate to predictions by others. I therefore request further revision and that the authors add a table and/or paragraph(s) in the Discussion that attempts to compare their results to similarly-motivated predictions by others. While I appreciate the challenges in comparing predictions, that is all the more reason to try to do it. If the authors determine it is impossible (which I do not think so), then why this so difficult is of great interest. I think the authors will find they can compare, at some level, their predictions to the predictions from some of the other studies. How they compare (even qualitatively), why they can be compared to some studies but not others, and ideas for making predictions more comparable will strengthen the manuscript."

- Thank you for this suggestion. A table has now been added to the manuscript (Table 6) which compares the results found in this study with those found in other recent studies of how climate change will impact Chesapeake Bay hypoxia. Additional text has been added to the discussion section referencing this table.
- Additionally, minor revisions and technical corrections have been completed following correspondence with the associate editor